# Decompose, Analyze and Rethink: Solving Intricate Problems with Human-like Reasoning Cycle

**Shangzi Xue**[1]    **Zhenya Huang**[1,2*]    **Jiayu Liu**[1]    **Xin lin**[1]    **Yuting Ning**[1]
**Binbin Jin**[1]    **Xin Li**[1]    **Qi Liu**[1,2]

1: State Key Laboratory of Cognitive Intelligence, University of Science and Technology of China
2: Institute of Artificial Intelligence, Hefei Comprehensive National Science Center
{xueshangzi,jy251198,linx,ningyt,bb0725}@mail.ustc.edu.cn;
{huangzhy,leexin,qiliuql}@ustc.edu.cn

## Abstract

In this paper, we introduce DeAR (*Decompose-Analyze-Rethink*), a framework that iteratively builds a reasoning tree to tackle intricate problems within a single large language model (LLM). Unlike approaches that extend or search for rationales, DeAR is featured by 1) adopting a tree-based question decomposition manner to plan the organization of rationales, which mimics the logical planning inherent in human cognition; 2) globally updating the rationales at each reasoning step through natural language feedback. Specifically, the *Decompose* stage decomposes the question into simpler sub-questions, storing them as new nodes; the *Analyze* stage generates and self-checks rationales for sub-questions at each node level; and the *Rethink* stage updates parent-node rationales based on feedback from their child nodes. By generating and updating the reasoning process from a more global perspective, DeAR constructs more adaptive and accurate logical structures for complex problems, facilitating timely error correction compared to rationale-extension and search-based approaches such as Tree-of-Thoughts (ToT) and Graph-of-Thoughts (GoT). We conduct extensive experiments on three reasoning benchmarks, including ScienceQA, StrategyQA, and GSM8K, which cover a variety of reasoning tasks, demonstrating that our approach significantly reduces logical errors and enhances performance across various LLMs. Furthermore, we validate that DeAR is an efficient method that achieves a superior trade-off between accuracy and reasoning time compared to ToT and GoT.

## 1 Introduction

Learning to perform intricate reasoning, including commonsense reasoning [23], knowledge reasoning [28], and mathematical reasoning [8], is a crucial step towards achieving general artificial intelligence [49, 20, 25, 26, 21, 24]. The tasks always present a significant challenge as they require many human-like intricate problem-solving abilities, such as abstract thinking and logical inference, which could consolidate many decision-making applications in real-world scenarios [38, 36, 15, 34, 53, 55].

Recent advances have witnessed remarkable performances of scaled-up large language models (LLMs) in various reasoning tasks, including GPT [5], LLaMA [40], and ChatGLM [9]. They could enable several state-of-the-art prompting approaches like Chain-of-Thought (CoT) [45], Tree-of-Thoughts (ToT) [49], Graph-of-Thoughts (GoT) [3], etc., to enhancing reasoning capabilities. They not only improve problem-solving performance but also reveal their intrinsic reasoning steps

---

*Corresponding Author

38th Conference on Neural Information Processing Systems (NeurIPS 2024).

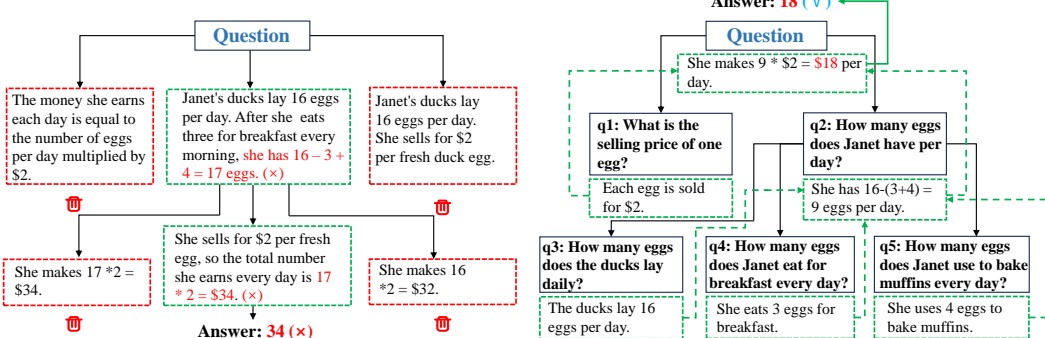

**Question:** Janet's ducks lay 16 eggs per day. She eats three for breakfast every morning and bakes muffins for her friends every day with four. She sells the remainder at the farmers' market daily for $2 per fresh duck egg. How much in dollars does she make every day at the farmers' market?

(a) The simulation of ToT Reasoning    (b) The simulation of DeAR Reasoning

Figure 1: Comparison between Tree-of-Thoughts (ToT) Reasoning and our DeAR (*Decompose-Analyze-Rethink*) Reasoning on a reasoning-based problem. (a) The simulation of Tree-of-Thoughts (ToT) (branch = 3). (b) The simulation of DeAR (*Decompose-Analyze-Rethink*) Reasoning.

(i.e., rationales) [47] through linear, tree-based, or graph-based structures. For example, in Figure 1 (a), given a math problem "Janet's ducks . . . in dollars . . . market?", ToT maintains a tree of thoughts with intermediate nodes to generate the rationales step by step. Specifically, through several operations including exploration, termination, and traceback on the nodes, ToT ultimately identifies the complete reasoning path, highlighting two-step rationales (green nodes) leading to the answer. However, although ToT and its variants [27, 35] perform the reasoning process explicitly, such a rationale-extension and search-based reasoning paradigm is still far from human-like intelligence and limits problem-solving abilities to some extent: On one hand, this tree-like structure is rigid and sometimes illogical. The ToT approaches often require setting a fixed number of thought branches ("3" branches in Figure 1 (a)) each time it expands, which can result in either missing information or redundancy. Its reasoning process essentially extends previous rationales at each step, but falls short of the logical planning inherent in human thinking [33, 42]. On the other hand, ToT generates rationale paths sequentially, and errors along the path, such as incorrectly calculating "she has 16-3+4=17 eggs", cannot be promptly corrected. This allows mistakes to propagate to subsequent steps, ultimately leading to an incorrect final outcome (e.g., "34").

To address these challenges, we propose a novel reasoning paradigm **DeAR (*Decompose-Analyze-Rethink*)**, which enhances LLMs' capacity for complex problem-solving by emulating human reasoning (Figure 1 (b)). This approach is inspired by several theories in cognitive science [43, 30]. Specifically, reasoning simplification theory [33] suggests that when confronted with an intricate question, humans tend to break it down into simpler ones, which help in organizing thoughts and solving problems more logically. Referring back to Figure 1 (b), we can break down the logic by first solving two sub-questions ($q_1$ and $q_2$). Upon examining $q_2$, we find it can be further divided into three additional sub-questions ($q_3$, $q_4$, and $q_5$). By sequentially resolving these sub-questions and using their results as feedback to update answers for previously generated sub-questions ($q_1$ and $q_2$), we ultimately arrive at the final answer ("18").

To implement such a human-like problem-solving process, we introduce a *Decompose-Analyze-Rethink* cycle. This involves gradually constructing a reasoning tree guided by sub-questions, following a top-to-bottom reasoning process as illustrated in Figure 1 (b). The process begins with the *Decompose* stage (black arrows in Figure 1 (b)), where a prompt-based method breaks down the question into simpler sub-questions at subsequent nodes. Then, the *Analyze* stage (green box at each node) takes charge of problem-solving at the node level. The stage also introduces a self-check module to ensure the quality of the generated rationales, thus refines the reasoning process. Last, in the *Rethink* stage (indicated by green arrows), the result at the current node is evaluated to determine if the reasoning in parent nodes requires further updates, providing a global perspective. After multiple cycles, the answer can be summarized from the root node.

Compared to ToTs [49, 27, 35] and GoT [3], our approach presents the following highlights. First, unlike ToT/GoT methods which directly generate rationales as branches from the original question,

DeAR breaks it into sub-question tree nodes to guide the generation. Second, our tree structure is more flexible and adaptable, as each node is generated and updated autonomously by the large language model based on the problem's logic, without relying on predefined settings. Third, DeAR enables timely correction of rationales, ultimately ensuring the correctness of the root node's answer.

We conduct extensive experiments on three complex reasoning benchmarks including ScienceQA [28], StrategyQA [12], and GSM8K [8]. Experimental results show that our approach enhances the reasoning performance with different backbones such as GPT-3.5 [1], LLaMA2 [40], and ChatGLM3 [9]. Compared to state-of-the-art methods such as Tree-of-Thoughts (ToT) and Graph-of-Thoughts (GoT), DeAR demonstrates a significant improvement in reasoning accuracy across all backbone LLMs, validating its generalizability and scalability. Additionally, by measuring the relationship between reasoning accuracy and reasoning time across different datasets, DeAR exhibits greater efficiency, further underscoring its advantages in practical applications.

## 2  Related Work

### 2.1  Prompt-based Approaches in LLM Reasoning

There has been a growing interest in LLM reasoning research, with various prompting schemes applied in areas such as commonsense [23], mathematical [8] and knowledge reasoning [29], etc. Early methods appends examples on top of the input question (few-shot prompting [5] or performs in-context learning (ICL) [37]), or includes no examples at all (zero-shot prompting) [44].

Recent research has sought to enhance the capabilities of large language models (LLMs) by introducing intermediate reasoning steps into the prompting process, epitomized by methods such as the Chain-of-Thought (CoT) [45]. By prompting LLMs to solve problems step by step, the CoT method demonstrates outstanding performance in multi-step reasoning tasks. Self-consistency [41] is a significant improvement upon CoT, where multiple CoT paths are initially generated, and the best one is selected as the final result, thereby improving the reliability of the outputs. In parallel, other prompting methods design search-based schemes for LLMs, such as Tree-of-Thoughts (ToT) [49] and Graph-of-Thoughts (GoT) [3] which innovate by structuring the reasoning process into tree or graph structures. These structures are created to take advantage of the many reasoning paths that LLMs can generate, greatly expanding the range and depth of exploration for any given question. More recently, Reasoning via Planning (RAP) [16] repurposes the LLM as both a world model and a reasoning agent to conduct reasoning. These methods expand the reasoning space of LLMs, which can fully leverage the diverse thinking paths generated by LLMs.

### 2.2  Question Decomposition

Question decomposition, which decomposes complex questions into multiple sub-ones, has been shown to largely improve models' reasoning ability. Early works [4] decompose questions with hand-crafted rules and lexicon-syntactic features. These works heavily rely on human efforts, which are hard to extend to general domains and tasks. Recently, researchers utilize neural network models to decompose questions [39, 18, 52]. For example, Min et al. [32] focused on directly training a model to produce sub-questions using question spans; BREAK [46] followed an alternative paradigm of collecting full question decomposition meaning representations (QDMR) annotations. However, a primary challenge lies in the scarcity of annotations for training a decomposition model [32].

More recently, in the era of LLMs, there are a lot of work exploring LLMs for question decomposition [50, 19, 17, 10, 7, 22, 51]. For example, ToT [49] prompts the LLM to decompose the rationales by searching intermediate steps. Least-to-most prompting [56] leverages a few examples to teach LLMs to decompose each problem into a series of simpler sub-problems. These prompting-based question decomposition methods serve as an important step in reasoning and planning with LLMs.

## 3  Problem Formulation and Preliminaries

### 3.1  Problem Definition

In this paper, we focus on the intricate reasoning task. The input of the task is the question $Q$ (e.g., "Janet's ducks ... market?" in Figure 1). The output is a rationale $R = (r_1, r_2, ..., r_k)$ with $k$ word

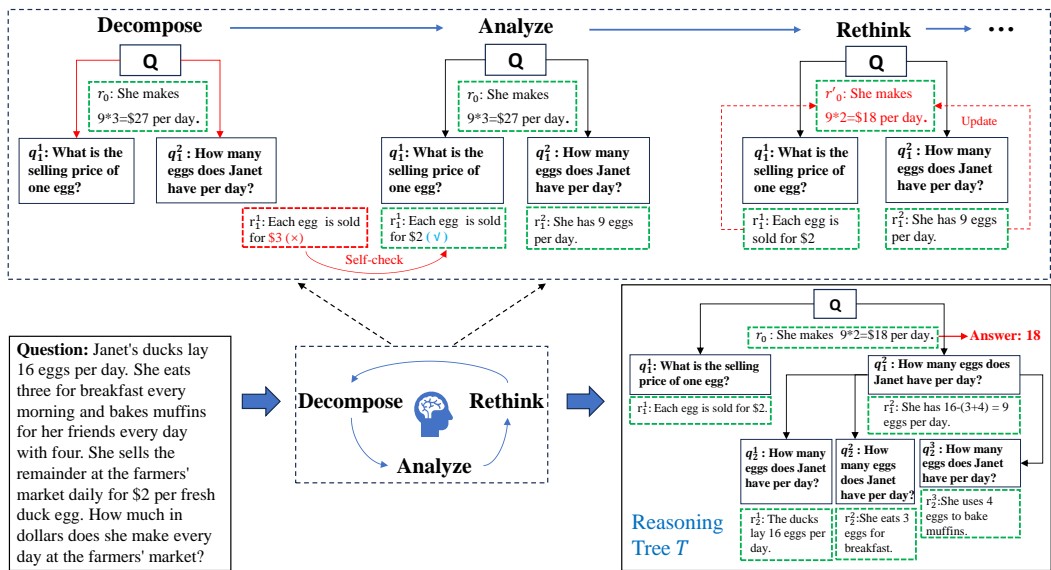

Figure 2: A demonstration of the DeAR (*Decompose-Analyze-Rethink*) cycle.

tokens ("She makes $9 \times \$2 = \$18$ per day."), and the answer $A$ ("18") derived from $R$. Given the input question $Q$, we aim to design a reasoning framework with LLM backbone $p_\theta$ to generate the rationale $R$ and answer $A$ as outputs.

## 3.2 Reasoning Tree

Motivated by the reasoning simplification theory [33], we propose a novel reasoning structure for LLMs, named Reasoning Tree $T$, as shown in Figure 1(b). Overall, this Reasoning Tree decomposes and resolves sub-questions using a top-down approach, while concurrently updating existing solutions through a bottom-up process. Formally, the Reasoning Tree $T$ can be defined as $T = (N, E)$ where $N$ is the set of tree nodes and $E$ is the edge set. Each node $n = (q, r, s) \in N$ contains a question $q$ as a sub-question of the target $Q$ (e.g., $q_2$ "How many eggs does Janet have per day?"), a rationale $r$ to $q$ ("She has 16-(3+4) = 9 eggs per day."), and a score $s$ evaluating the logical coherence of $r$. Each directed edge $e = (n_p, n_c) \in E$ means that the upper-level sub-question $q_p$ in the parent node $n_p$ is decomposed into a lower-level one $q_c$ in the child node $n_c$ (e.g., the parent $q_2$ "How many eggs ... have per day" is decomposed into three children $q_3$ "How many eggs ... lay", $q_4$ "How many eggs ... breakfast", and $q_5$ "How many eggs ... muffins").

Our Reasoning Tree is progressively constructed and updated. The target question $Q$ in the root node is decomposed into sub-questions step by step, from sub-questions in the higher levels to the ones in the lower levels (i.e., the black directed edges in Figure 1). For example, $Q$ is first decomposed into $q_1$ and $q_2$, then $q_2$ is further decomposed into $q_3$, $q_4$ and $q_5$. Furthermore, humans could also rethink the rationales generated earlier (in the higher nodes) based on the ones generated later (in the lower nodes). For example, the rationales for $q_4$ ("She eats 3 eggs for breakfast") could be used to update rationales for $q_2$ ("She has 16-(3+4) = 9 eggs per day") through the dashed lines in green.

## 3.3 Framework Overview

To construct the aforementioned Reasoning Tree $T$, which imitates human-like reasoning, we propose a novel **DeAR (*Decompose-Analyze-Rethink*)** cycle as the core of our framework, as illustrated in Figure 2. The cycle is composed of three stages: *Decompose*, *Analyze* and *Rethink*. Specifically, in the *Decompose* stage, one upper-level question is decomposed into several lower-level ones. In the *Analyze* stage, the framework solves the newly generated sub-questions by generating and self-checking rationales. In the *Rethink* stage, the newly generated rationales are used to update existing ones in the parent nodes. The three stages work in a cycle to build the reasoning tree $T$.

# 4 DeAR (*Decompose-Analyze-Rethink*) Cycle

In this section, we will demonstrate how the reasoning tree $T$ is constructed with the *Decompose-Analyze-Rethink* cycle, as demonstrated in Figure 2.

Initially, the target question $Q$ is set as the question $q_0$ in the root node $n_0$. The framework selects an existing edge node $n_t = (q_t, r_t, s_t)$ ($t$ is the level of the node) from $T$ (e.g., $n_0$ with $Q$ "Janet's ducks ... market?") to start the cycle. First, in the *Decompose* stage (4.1), we prompt LLMs to decompose the question $q_t$ in the node into sub-questions $q_{t+1}$ if possible, and store them in nodes $n_{t+1}$ at level $t + 1$ (e.g., $q_1^1$ "What is ... one egg?", and $q_1^2$ "How many ... per day?"). Then, in the *Analyze* stage (4.2), we conduct reasoning and answers the newly generated questions $q_{t+1}$ by generating rationales $r_{t+1}$ for them ($r_1^1$ "Each egg is sold for \$2" for $q_1^1$, and $r_1^2$ "She has 16 eggs per day" for $q_1^2$), checking their correctness and evaluating the coherence scores $s_{t+1}$ (Eq. (5)). Next, in the *Rethink* stage (4.3), we use the newly generated $r_{t+1}$ to update rationales in existing upper-level nodes $r_i(i \leq t)$ (e.g., use $r_1^1$ and $r_1^2$ to update $r_0$ into $r_0'$). After that, the framework selects another edge node and returns to the *Decompose* stage (e.g., decompose $q_1^2$ into $q_2^1$, $q_2^2$ and $q_2^3$). The cycle continues until the LLMs determine that no further decomposition is possible, thereby forming the reasoning tree $T$ for $Q$.

As $Q$ is the question $q_0$ for the root node $n_0$, after the tree-construction process, we consider the rationale $r_0$ in the root node as the overall solution for $Q$ and extract the answer $A$ from $r_0$. The whole procedure is described in Algorithm 1. In the following sections, we will technically describe the three stages in the cycle and make detailed analyses.

## 4.1 *Decompose* Stage

According to the Analogical Reasoning theory [2], when humans conduct reasoning, they often analogize the logical processes of new questions to those of similar questions. Therefore, to make the decomposition logic of sub-questions $q_t$ at each level $t$ more closely resemble that of humans, we first use human-annotated question decomposition examples (*Appendix* A.1) as a demonstration pool $P$. Then we calculate the cosine similarity of the representations between $Q$ and each $Q_i^d$ in $P$ and select top-$K$ nearest neighbors in the vector space. After that, we concatenate each $Q_i^d$ with its human-annotated sub-questions $subqs^i = (subq_1^i, subq_2^i, ..., subq_n^i)$ to form $K$ question-decomposition examples (*Appendix* A.1)

$$lh_Q = (Q_i^d, subqs^i)(i = 1, 2, ..., K). \quad (1)$$

These examples are regarded as "logic heuristics" that inspire the model to decompose questions in a manner closely aligned with human reasoning.

After obtaining $lh_Q$, we utilize them to decompose the sub-question $q_t$ at level $t$ into multiple sub-questions at level $t + 1$. Specifically, given question $q_t$, if its coherence score $s_t$ (Eq. (5)) is higher than a threshold $\epsilon_1$, We ask the LLM whether it needs to be further decomposed. If $q_t$ requires decomposition, we then prompt the LLM to autonomously break it down into several sub-questions $\{q_{t+1}^j, j = 1, ..., J\}$. It is worth noting that in our decomposition approach, we do not pre-specify the number $J$ of sub-questions; instead, we allow LLMs to adap-

---

**Algorithm 1** *Decompose-Analyze-Rethink*

---

**Input:** Question $Q$
**Parameters:** LLM $p_\theta$, natural language prompts ($c_1 \sim c_6$), threshold $\epsilon_1$ for *Decompose*, threshold $\epsilon_2$ for *Rethink*
**Output:** Rationale $R$, Answer $A$
Create an empty node queue $N$
Enqueue $n_0(q_0 = Q, r_0 = None, s_0 = 1)$ into $N$
**while** $N$ is not empty and current level < max depth[a]
**do**
    Dequeue current node $n_t(q_t, r_t, s_t)$ from $N$
    **if** $n_t$ is an end node $n_{end}$ **then**
        continue
    **else if** $s_t > \epsilon_1$ **then**
        *// Stage 1: Decompose*
        $\{q_{t+1}^j\} \leftarrow Decompose(p_\theta, h_1, lh_Q, q_t)$ (2)
        *// Stage 2: Analyze*
        $r_{t+1}^j \leftarrow Solve(p_\theta, h_2, q_{t+1}^j)$ (3)
        $\hat{r}_{t+1}^j \leftarrow Self\_Check(p_\theta, h_3, q_{t+1}^j, r_{t+1}^j)$ (4)
        $s_{t+1}^j \leftarrow Score(p_\theta, h_4, q_{t+1}^j, \hat{r}_{t+1}^j)$ (5)
        Set $n_{t+1}^j \leftarrow (q_{t+1}^j, \hat{r}_{t+1}^j, s_{t+1}^j)$ (6)
        Enqueue $n_{t+1}^j$ into $N$
        *// Stage 3: Rethink*
        **if** $s_{t+1}^j > \epsilon_2$ **then**
            $L_k \leftarrow Extract(p_\theta, h_5, L, q_{t+1}^j)$ (7)
            $r' \leftarrow Update(p_\theta, h_6, n_e(q, r, s), \hat{r}_{t+1}^j)$ (8)
            $n_e(q, r', s) \leftarrow n_e(q, r, s)$ (6)
    **else**
        Enqueue $n_{end}$ into $N$
    **end if**
**end while**
$R \leftarrow r_0$
Extract answer $A$ from $R$
**return** $R, A$

---

[a]See 5.1.2 for max depth and branch settings.

---

tively determine it based on the logic of each question. However, the number of sub-questions is capped at a predefined maximum branch limit to ensure computational efficiency and manageability 5.1.2. This enhances adaptability and more closely aligns with human logical characteristics when compared to existing methods like ToT [49] and GoT [3], etc. To facilitate this process, we design a heuristic-enhanced prompt that consists of a prompt head $h_1$ and "logic heuristics" $lh_Q$. The prompt head describes the question decomposition task in natural language. This process is formulated in Eq. (2). Additionally, we validate the effectiveness of using logic heuristics, and provide detailed explanations and templates in *Appendix* A.1.

$$\{q_{t+1}^j, j = 1, ..., J\} \leftarrow Decompose(p_\theta, h_1, lh_Q, q_t). \tag{2}$$

After decomposition, each $q_{t+1}^j$ is added as a new node $n_{t+1}^j$ at level $t + 1$, with a directed edge from $n_t$ to $n_{t+1}^j$ (denoted as $e^j = (n_t, n_{t+1}^j)$). If the LLM determines that $q_t$ does not require further decomposition, we create a leaf node $n_{end}$ as a child of $n_t$.

### 4.2 *Analyze* Stage

In *Analyze* stage, we reason the answers for all the sub-questions $\{q_{t+1}^j\}$ at level $t + 1$. To be specific, we first prompt the LLM to generate the essential rationale $r_{t+1}^j$ for each sub-question $q_{t+1}^j$:

$$r_{t+1}^j \leftarrow Solve(p_\theta, h_2, q_{t+1}^j). \tag{3}$$

Here, $h_2$ denotes the prompt head, which is a natural language sentence that asks the model to generate detailed solutions (see *Appendix* A.2).

After obtaining the rationales for the sub-questions, we evaluate and correct them, as large language models (LLMs) often tend to hallucinate during problem-solving [54]. Using generated rationales without verification can propagate errors, leading to incorrect outcomes. To address this issue, we develop a self-check method that promptly identifies and corrects these errors while providing a coherence score (Eq. (5)) for each node.

Specifically, we first instruct the LLM to perform a self-check on the rationale $r_{t+1}^j$ generated for the sub-question $q_{t+1}^j$ (see *Appendix* A.2 for the prompt head $h_3$) to identify any potential errors. If the LLM detects errors in the original rationale $r_{t+1}^j$, it modifies the rationale to $\hat{r}_{t+1}^j$; otherwise, the rationale is output unchanged. Take the case in Figure 2 as an example, we expect the LLM to identify the error "Each egg is sold for \$3" in $r_1^1$, and correct it to "Each egg is sold for \$2". This process is denoted as:

$$\hat{r}_{t+1}^j \leftarrow Self\_Check(p_\theta, h_3, q_{t+1}^j, r_{t+1}^j). \tag{4}$$

Then, we prompt the LLM to evaluate the logical coherence between the refined rationale $\hat{r}_{t+1}^j$ and the question $q_{t+1}^j$, by generating a coherence score $s_{t+1}^j$ (see *Appendix* A.2 for prompt head $h_4$):

$$s_{t+1}^j \leftarrow Score(p_\theta, h_4, q_{t+1}^j, \hat{r}_{t+1}^j). \tag{5}$$

The score $s_{t+1}^j$ can also be obtained through voting or classification methods. Here, we specifically investigate the effectiveness of directly prompting LLMs to generate numerical values as scores.

At the end of the *Analyze* stage, we fill the obtained rationales and scores into nodes $n_{t+1}^j (j \geq 1)$:

$$n_{t+1}^j = (q_{t+1}^j, \hat{r}_{t+1}^j, s_{t+1}^j). \tag{6}$$

where $s_{t+1}^j$ can support the current or subsequent cycles in *Rethink* (4.3) and *Decompose* (4.1).

### 4.3 *Rethink* Stage

According to self-reflection theories [11, 13, 6] in cognitive science, humans constantly update and reflect on their previous reasoning results based on the current information. This allows us to correct past mistakes and ultimately achieve a consistent and stable answer. For example in Figure 2, a person might initially answer question $Q$ ("Janet's ducks ... How much ... market?") with the rationale $r_0$ "She makes $9 \times 3 = \$27$ per day". However, after considering responses to sub-questions $q_1^1$ ("What

is the selling price of one egg?") and $q_1^2$ ("How many eggs does Janet have per day?"), he/she realizes an error in $r_0$. The correct calculation, using the values "2" for the price per egg and "9" for the daily number of eggs, should be "$2 \times 9 = \$18$".

Nevertheless, existing methods like ToT [48] search reasoning paths based solely on preceding steps, lacking the ability to retrospectively update earlier content based on the influence of later steps. To address this, we introduce a *Rethink* stage that mirrors the human reflective process.

Specifically, during the rethinking process, humans first identify which existing reasoning steps may require revision. We aim to automate this by using LLMs to detect logical connections between ancestral and newly generated nodes, updating ancestral nodes based on insights from the rationales of new nodes. In our proposed "Reasoning Tree", we essentially use information from lower-level nodes to "rethink" higher-level nodes, closely mirroring the human cognitive simplification process in problem-solving [33].

To achieve this, after obtaining node $n_{t+1}^j$ in *Analyze* Stage, we first check its coherence score $s_{t+1}^j$ (Eq. (5)). If $s_{t+1}^j$ exceeds the threshold $\epsilon_2$, we then examine the correlation between $q_{t+1}^j$ and all sub-questions above level $t$, specifically, $\{q_l, \ l \leq t\}$. Next, we extract a subset of $k$ most related nodes $L_k$ from $L \triangleq \{n_l, \ l \leq t\}$ (the specific nodes to be extracted are determined by the LLM):

$$L_k \leftarrow Extract(p_\theta, \ h_5, \ L, q_{t+1}^j), \ L_k \subseteq L. \tag{7}$$

where $h_5$ is a prompt head (*Appendix* A.3). Next, we use the rationale $\hat{r}_{t+1}^j$ of sub-question $q_{t+1}^j$ to update the rationale $r$ of each extracted node $n_e$ in $L_k$:

$$r' \leftarrow Update(p_\theta, \ h_6, \ n_e(q, r, s), \ \hat{r}_{t+1}^j). \tag{8}$$

Finally, we replace $r$ with the updated rationale $r'$:

$$n_e(q, r', s) \leftarrow n_e(q, r, s). \tag{9}$$

# 5 Experiments

In this section, we demonstrate the generality and effectiveness of DeAR by applying it to a wide range of tasks, including knowledge reasoning, logical reasoning and mathematical reasoning. The results across these tasks validate DeAR's adaptability and highlight its capability to effectively tackle a diverse range of challenging reasoning tasks.

## 5.1 Experimental Setup

### 5.1.1 Datasets and Baselines

We employ the **ScienceQA** [28] dataset for the knowledge reasoning task. And we use **StrategyQA** [12] for logical reasoning that requires multiple reasoning steps. We also verify the mathematical reasoning ability of our framework by applying it to **GSM8K** dataset [8]. The details of these datasets are available in *Appendix* B.1.1.

In our main results, we compare DeAR with multiple prompt-based methods including **Few-shot** prompting [5], **Chain-of-Thoughts (CoT)** prompting [45], and state-of-the-art **Tree-of-Thoughts (ToT)** [49] and **Graph-of-Thoughts (GoT)** [3] prompting. Besides, we also list extra comparison results with another two state-of-the-art prompt-based methods **Least-to-most** Prompting [56] and **SelfCheck** [31] (see *Appendix* B.1.2 for all baseline details).

### 5.1.2 Implementation Details

We conduct experiments with three LLM backbones **GPT-3.5** [1], **LLaMA2-7B** [40] and **ChatGLM3-6B** [9]. For GPT-3.5, we use the OpenAI API to invoke the "gpt-3.5-turbo-1106" model. For LLaMA2-7B and ChatGLM3-6B, we load the checkpoints from huggingface[2][3] and use the models directly

---

[2]https://huggingface.co/THUDM/chatglm3-6b
[3]https://huggingface.co/meta-llama/Llama-2-7b

Table 1: Overall results of our DeAR Framework on three intricate reasoning datasets. ($* : p < 0.05$).

| | ScienceQA | | | StrategyQA | | | GSM8K | | |
|---|---|---|---|---|---|---|---|---|---|
| | GPT-3.5 | LLaMA2 | ChatGLM3 | GPT-3.5 | LLaMA2 | ChatGLM3 | GPT-3.5 | LLaMA2 | ChatGLM3 |
| Few-shot | 73.97 | 66.35 | 42.46 | 67.71 | 61.21 | 54.41 | 74.26 | 72.25 | 51.02 |
| CoT | 75.17 | 67.58 | 46.35 | 69.26 | 63.86 | 57.18 | 79.55 | 74.04 | 53.85 |
| ToT | 82.52 | 69.01 | 49.58 | 71.89 | 66.52 | 59.21 | 83.42 | 75.22 | 55.88 |
| GoT | 82.34 | 68.86 | 49.26 | 72.02 | 66.61 | 59.88 | 84.77 | 75.95 | 56.01 |
| **DeAR** | **83.68*** | **70.57*** | **51.08*** | **73.36*** | **68.33*** | **61.02*** | **86.82*** | **78.01*** | **58.54*** |
| Least-to-most | 76.61 | 68.02 | 47.45 | 70.55 | 64.43 | 58.36 | 81.25 | 74.67 | 54.21 |
| SelfCheck | 75.81 | 69.33 | 49.23 | 68.87 | 66.35 | 61.22 | 79.88 | 75.28 | 56.72 |

without fine-tuning as the backbone.[4] For each dataset, we randomly sample 10% of its training set as a validation set to select different combinations of thresholds $\epsilon_1$ and $\epsilon_2$. The combination that achieves the best performance on the validation set is then used for inference on the test set. We observe that the threshold combinations obtained through this method also yield optimal inference results on the test set. In Section 5.6, we visualize the inference accuracy on the test sets across different datasets based on GPT-3.5, using diffenrent threshold combinations. The implementation and prompting templates (i.e., natural language prompts $h_1 \sim h_6$ for *Decompose*, *Analyze* and *Rethink* ) are shown in *Appendix* A. To ensure computational efficiency, we set the maximum depth to 4 and the maximum number of branches to 3 during the construction of the reasoning tree in DeAR. This prevents the tree from becoming excessively deep and avoids redundancy in sub-questions. For baselines, the settings used in the experiments are consistent with those described in the original papers. For a concise description of baselines, please refer to *Appendix* B.1.2.

## 5.2 Experimental Results

We conduct experiments to verify the effectiveness of our framework DeAR, and report the results in Table 1. We use the accuracy (ACC) as metric for all three datasets. We statistically test the improvement over baselines with paired t-test, and find the improvement to be significant with $p < 0.05$ (marked with "$*$"). We get the following observations. First, DeAR performs better than all baselines, which indicates it is more effective in enhancing LLMs' reasoning ability. Second, the improvements over ToT highlight the advantage of *Decompose* stage which adaptively decomposes questions based on their characteristics rather than extending a fixed number of thought branches. Third, DeAR performs better than GoT which lacks rationale updating. This reflects the superiority of the *Rethink* stage to identify correlations between reasoning steps and update previous rationales. Besides, the accuracy increase on GSM8K is greater than ScienceQA and StrategyQA. That is probably because problems in GSM8K require longer rationales to be solved (Table 2). Furthermore, DeAR outperforms the Least-to-most [56] and SelfCheck [31] methods across all datasets. The Least-to-most method sequentially solves sub-problems derived from the decomposition without updating content that has already been generated; SelfCheck updates rationales but it does not decompose the original question. In contrast, DeAR not only generates rationales based on decomposed sub-questions but also updates existing rationales in each cycle. This further underscores the necessity of the Decompose and Rethink phase in DeAR for enhancing the reasoning capabilities of LLMs.

We have also validated that DeAR enhances stronger LLMs (e.g., GPT-4) on complex reasoning tasks (e.g., MATH), as shown in *Appendix*. *Appendix* B.3 includes an ablation study on the self-check method in the *Analyze* stage, as its removal does not structurally impact the other stages.

## 5.3 Analyses of the Reasoning Tree

For each question $Q$, DeAR constructs a reasoning tree $T$ to represent the reasoning process, as shown in Figure 1 (b). The structure of $T$ provides insights into the complexity of $Q$. To analyze the nature of questions across datasets, we examine reasoning trees from three datasets

Table 2: Characteristics of $T$ in different datasets.

| | ScienceQA | StrategyQA | GSM8K |
|---|---|---|---|
| Avg Branch | 1.58 | 2.43 | 2.06 |
| Avg Depth | 3.62 | 1.96 | 2.55 |
| Avg Length of $R$ | 66.34 | 61.55 | 85.27 |

using three metrics: "Avg Branch," "Avg Depth," and "Avg Length of $R$." "Avg Branch" indicates the average branching factor of $T$, "Avg Depth" reflects the average depth of $T$, and "Avg Length

---

[4]Our code is available at: `https://github.com/ShangziXue/DeAR`

Table 3: ROSCOE evaluation results of rationales generated by Tree-of-Thoughts (ToT), Graph-of-Thoughts (GoT) and DeAR on different datasets. SC = Source-Consistency; RA = Reasoning Alignment.

|  | ScienceQA | | StrategyQA | | GSM8K | |
|---|---|---|---|---|---|---|
|  | SC | RA | SC | RA | SC | RA |
| ToT | 0.44 | 0.31 | 0.47 | 0.33 | 0.56 | 0.41 |
| GoT | 0.42 | 0.35 | 0.44 | 0.38 | 0.53 | 0.45 |
| **DeAR** | **0.48** | **0.42** | **0.52** | **0.43** | **0.58** | **0.50** |

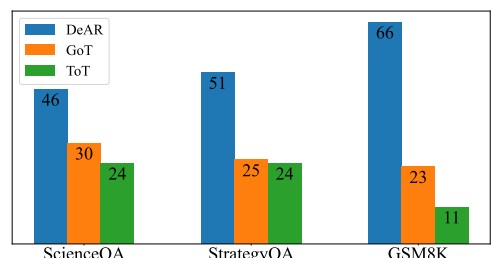

Figure 3: The distributions of annotators' selections. More annotators considered DeAR's rationales to be more logical.

of $R$" represents the length of rationale $R$ derived from the root node $n_0$ upon tree completion, e.g., $R = r_0$:"She makes 9*2=\$18 per day" in Figure 2.

Using GPT-3.5 as the backbone, results in Table 2 reveal the following: ScienceQA questions have the highest "Avg Depth" and lowest "Avg Branch," indicating fewer sub-questions per *Decompose* stage but more rounds required. StrategyQA questions have the lowest "Avg Branch" but the highest "Avg Depth," suggesting fewer *Decompose* rounds but more sub-questions per round. For GSM8K, the root node $n_0$ has longer rationales $R$, suggesting that these questions require more extensive explanations than those in the other datasets.

### 5.4   Logical Coherence of the Generated Rationales

We assess the logical coherence of rationales generated by DeAR using both automatic and human evaluation methods. For automatic metrics, we apply the Source-Consistency" (SC) and Reasoning Alignment" (RA) from the ROSCOE evaluation suite [14]. SC measures logical entailment between question and rationale, while RA evaluates alignment with ground truth. As shown in Table 3, DeAR outperforms ToT and GoT on all datasets. For human evaluation, 100 questions were sampled from each dataset, with annotators selecting the most logical rationale among those generated by ToT, GoT, and our method (details in *Appendix* B.4). Results in Figure 3 confirm that DeAR (using GPT-3.5) produces rationales with superior logical coherence compared to ToT and GoT.

### 5.5   Effectiveness of Rethink

In *Rethink* stage, our DeAR employs the same backbone LLMs to determine which nodes' rationales need to be updated. To validate its effectiveness, based on GPT-3.5, we compare our method with "Random Update" method which randomly selects nodes to update at different proportions. The results in Table 3 demonstrate

Table 4: Comparisons between different portions of "Random Update" and DeAR.

| Random Update | ScienceQA | StrategyQA | GSM8K |
|---|---|---|---|
| 0% | 82.77 | 72.84 | 85.09 |
| 20% | 81.77 | 72.21 | 83.96 |
| 40% | 82.59 | 73.03 | 84.35 |
| 60% | 82.06 | 72.29 | 85.07 |
| 80% | 81.49 | 72.04 | 86.01 |
| 100% | 81.16 | 71.79 | 85.32 |
| **DeAR** | **83.68** | **73.36** | **86.82** |

that, compared to "Random Update", our method performs better in terms of accuracy. Additionally, unlike approaches that require a 100% update of all generated rationales, DeAR's targeted updates allow the model to autonomously select nodes that need refinement, thus minimizing unnecessary inference.

### 5.6   Combinations of Thresholds

In this subsection, we visualize the impact of different combinations of threshold values $\epsilon_1$ and $\epsilon_2$ on the inference accuracy of DeAR (with GPT-3.5 backbone) across the test sets of all three datasets. $\epsilon_1$ and $\epsilon_2$ are set for the *Decompose* stage (Section 4.1) and *Rethink* stage (Section 4.3), respectively, with their value combinations selected based on

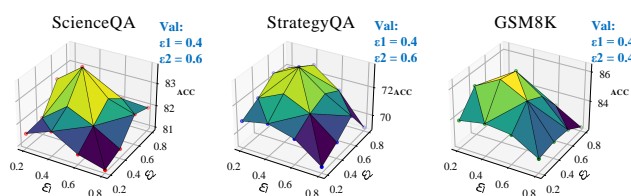

Figure 4: Combinations of threshold values ($\epsilon_1$, $\epsilon_2$) and corresponding ACCs on test sets (GPT-3.5 backbone).

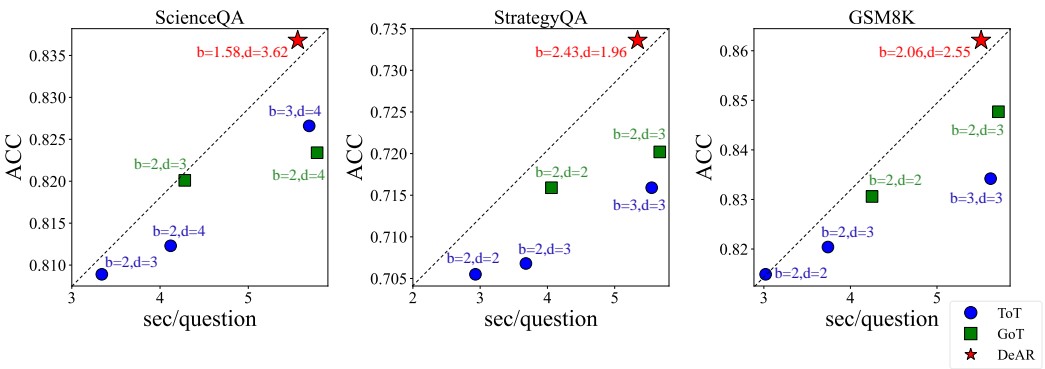

Figure 5: Efficiency comparison between DeAR and variants of ToT/GoT.

performance on the validation set (Section 5.1.2). We observe from Figure 4 that, DeAR achieves the highest accuracy when setting $\epsilon_1 = 0.4$ and $\epsilon_2 = 0.6$ for ScienceQA and StrategyQA. For GSM8K, the highest accuracy is obtained with $\epsilon_1 = 0.4$ and $\epsilon_2 = 0.4$. The threshold combinations that optimize DeAR's performance on the test set are consistent with those obtained from the validation set (e.g., Val: $\epsilon_1 = 0.4$; $\epsilon_2 = 0.6$ for ScienceQA), demonstrating the validity of the value selection method. Additionally, the smaller optimal $\epsilon_2$ value for GSM8K suggests that tackling GSM8K problems requires a more frequent or active rethinking process compared to ScienceQA and StrategyQA. This difference highlights the varying nature of reasoning demands across different tasks, where the threshold tuning helps adapt DeAR's reasoning process accordingly.

## 5.7 Efficiency

Compared to the rationale extension in ToT and GoT, DeAR incorporates question decomposition and rationale updating. Thus, will the efficiency of reasoning be affected? To investigate this, we use ChatGLM3-6B as the backbone model and measure the average inference time per question (seconds/question) and accuracy (ACC) for each method. The results are in the form of scattered points as shown in Figure 5. We set the fixed branch numbers and depths for these variants of ToT and GoT (e.g., b=3, d=4), and compare them with DeAR. In ToT/GoT, we set "b" and "d" (integers) as close to DeAR's average values as possible to ensure fairness. We can observe that points closer to the upper-left corner, and farther away vertically from the diagonal, represent methods that achieve a better trade-off between reasoning accuracy and time. The points corresponding to DeAR clearly exhibit this characteristic, hence we can conclude that it has higher efficiency. Moreover, in Appendix B.5, to further validate this conclusion, we measured the average number of API calls made by DeAR, ToT, and GoT per question in the ScienceQA dataset using GPT-3.5, as well as their reasoning accuracy. DeAR consistently requires fewer API calls on average to solve a question, while simultaneously achieving higher accuracy.

## 6 Conclusion

In this paper, we introduced DeAR (*Decompose-Analyze-Rethink*), a framework designed to mimic human reasoning patterns in tackling intricate problems by constructing a reasoning tree in a top-down, iterative manner. This approach is coupled with a *Decompose-Analyze-Rethink* cycle, in which the rationale at each node is generated, evaluated, and refined through feedback loops. Specifically, the *Decompose* stage applies logic heuristics to decompose the original question, the *Analyze* stage produces and self-checks rationales, and the *Rethink* stage integrates these insights by updating parent nodes based on child-node feedback. Extensive experiments demonstrate that DeAR not only improves reasoning performance across different large language models (LLMs) but also surpasses current state-of-the-art methods like Tree-of-Thoughts (ToT) and Graph-of-Thoughts (GoT) in logical coherence and accuracy. DeAR's rationale update mechanism enhances logical consistency by iteratively refining previously generated rationales, achieving more accurate and interpretable results. Additionally, compared to ToT and GoT, DeAR strikes a better balance between reasoning accuracy and inference time, further improving efficiency. Case studies also demonstrate that DeAR produces more interpretable reasoning process (*Appendix* B.6).

## Acknowledgments and Disclosure of Funding

This research was partially supported by grants from the National Natural Science Foundation of China (No.62106244, 62337001), and the Key Technologies R&D Program of Anhui Province (No.202423k09020039).

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

# A More Details and Prompt Templates of DeAR

## A.1 Prompts for *Decompose* Stage

Table 5: An example of Heuristic-enhanced Prompt $c_1$ in the Decomposition Stage.

| Example A.1: Prompts for *Decompose* Stage |
| --- |
| **Prompt Head** $h_1$: Your task is to decompose the given question $Q$ into sub-questions. You should based on the specific logic of the question to determine the number of sub-questions and output them sequentially. If you consider the question $Q$ to be sufficiently simple and no further decomposition is needed, then output "End." I will provide you with three questions similar to q, along with their decomposed sub-questions as examples. You can learn from these examples on how to decompose such questions, and then apply what you've learned to decompose $Q$. |
| **Logic Heuristics** $lh_Q$: 

 **Example question 1:** Will Queen Elizabeth be buried in the Pantheon? 
 **Decomposition:** (1): The Panthéon is reserved as a mausoleum for citizens of which country? (2): Is Queen Elizabeth from (1)? 

 **Example question 2:** Was Elizabeth II the Queen during the Persian Gulf War? 
 **Decomposition:** (1): When did Elizabeth II become the Queen? (2): When was the Persian Gulf War? (3): Was Elizabeth II alive in (2)? (4): Is (2) after (1)? (5): Are the answers to (3) and (4) both yes? 

 **Example question 3:** Does Elizabeth II reign over the Balearic Islands? 
 **Decomposition:** (1): What are all the areas Queen Elizabeth II rules over? (2): What country owns the Balearic Islands? (3): Is (2) included in (1)? |
| **The given question** $Q$ : Does the actress who played Elizabeth II speak fluent Arabic? 
 **Please note that:** If $Q$ can be decomposed, you should output multiple sub-questions as shown in the above Logic Heuristics. Otherwise please output "End". 
 **The decomposed sub-questions for** $Q$ **is:** |

In the *Decompose* stage (4.1), we design a heuristic-enhanced prompt to facilitate the question decomposition process. The prompt consists of a prompt head $h_1$ and "logic heuristics" $lh_Q$. The prompt head $h_1$ describes the question decomposition task in natural language and "logic heuristics" $lh_Q$ are $K$ ($K = 3$ in this paper) demonstrations of how similar questions are decomposed.

For the $lh_Q$, we specifically outline the process for obtaining these. We begin by constructing the question decomposition demonstration pool $P$. For GSM8K and StrategyQA, we directly utilize the existing question decomposition annotations from the training sets as $P$. In the case of ScienceQA, we generate question decomposition data from a portion of its training set using annotations produced by GPT-4, which are subsequently verified through manual checks. Next we employ a BERT encoder $E_\xi$ to transform target question $Q$ and questions $Q_i^d (i = 1, 2, ...M)$ from decomposition pool $P$ into vector representations $z_i$ and $z$, respectively. Then we calculate the cosine similarity of the representations between $Q$ and $Q_i^d$, and select top-$K$ nearest neighbors in the vector space:

$$I_d = argTopK \frac{z^T z_i}{\|z\|\|z_i\|}, (i = 1, 2, ..., M). \tag{10}$$

where $I_d$ is an index set of the top-$K$ similar questions of $Q$ in the demonstration pool. Finally, We concatenate each $Q_i^d (i \in I_d)$ with its human-annotated sub-questions $subqs^i = (subq_1^i, subq_2^i, ..., subq_n^i)$ to form $K$ question-decomposition examples.

$$lh_Q = (Q_i^d, subqs^i)(i = 1, 2, ..., K). \tag{11}$$

In this paper, we use the SentenceTransformers[5] as $E_\xi$ to transform questions into embeddings, and set $K = 3$.

---

[5] https://www.sbert.net/

Table 5 displays the prompt used in the Decomposition stage for a particular question in StrategyQA dataset.

We also conducted experiments demonstrating that incorporating logic heuristics effectively enhances the overall performance of the *Decompose* Stage. As shown in the table 6, on the ScienceQA dataset, DeAR w/o logic heuristics indicates the removal of logic heuristics from the prompts used in the *Decompose* Stage. This adjustment results in lower ACC compared to DeAR, highlighting the necessity of constructing heuristic-enhanced prompts for improved performance.

Table 6: Performance comparison of DeAR with and without logic heuristics on the ScienceQA dataset.

|  | GPT-3.5 | LLaMA2-7B | ChatGLM3-6B |
| --- | --- | --- | --- |
| DeAR w/o logic heuristics | 83.06 | 69.85 | 50.17 |
| DeAR | **83.68** | **70.57** | **51.08** |

## A.2 Prompts for *Analyze* Stage

For the generated sub-questions, in the *Analyze* stage 4.2, we use the $Solve$ method to prompt the LLM to generate rationales for them. The prompt used for this purpose is $h_2$. For $Self\_Check$, we use $h_3$ to correct the errors in the generated rationales. For $Score$, we use $h_4$ to prompt the LLM to score the logical coherence of rationales. Examples in Table 7 demonstrate $h_2$, $h_3$ and $h_4$.

Table 7: Demonstrations of $h_2$, $h_3$ and $h_4$ in the *Analyze* Stage.

| Example A.2: Prompts for *Analyze* Stage |
| --- |
| $Solve$ **prompt:** 
 $h_2$: Answer the following question and provide a detailed reasoning process. 

 Question: How many eggs does Janet have per day? 

 Your reasoning process: |
| $Self\_Check$ **prompt**: 
 $h_3$: There might be some errors in the rationale for the following question. If you believe there are errors, please correct them and provide the accurate reasoning process. Otherwise, output the original reasoning process. 

 Question: How many eggs does Janet have per day? 
 Rationale: Janet has 16 eggs per day. 

 Your output: |
| $Score$ **prompt**: 
 $h_4$: Please rate the overall correctness and logic of the following rationale on a scale from 1 to 10, where 1 is the lowest score and 10 is the highest score. Divide the chosen integer by 10 and output it as the final score. 

 Rationale: Janet has 16 eggs per day. 

 Your score: |

## A.3 Prompts for *Rethink* Stage

In the Rethink Stage 4.3, we first extract previous sub-questions that are relevant to the newly generated one by using the prompt $c_5$, then we use the newly generated rationale to update rationales of these previous sub-questions by using $c_6$. Examples are shown in Table 8.

Table 8: Demonstrations of $h_5$ and $h_6$ in the Rethink Stage.

| |
|---|
| **Example A.3: Prompts for *Rethink* Stage** |
| *Extract* **prompt:** |
| $h_5$: Please extract questions from the following list that might use the answer of $q$ to update their rationales. |
| |
| Question list: |
| 1. What is the selling price of one egg? |
| 2. How many eggs does Janet have per day? |
| |
| $q$: How many eggs does the ducks lay daily? |
| The answer of $q$: The ducks lay 16 eggs per day. |
| |
| Your extracted questions: |
| *Update* **prompt:** |
| $h_6$: Please update the answer to question b based on the answer to question a. |
| |
| Question a: How many eggs does Janet eat for breakfast every day? |
| The answer to question a: She eats 3 eggs for breakfast. |
| Question b: How many eggs does Janet have per day? |
| The answer to question b: She has 16 eggs per day. |
| |
| The updated answer to question b is: |

# B    Appendix for Experiments

## B.1    Datasets and Baselines

### B.1.1    Datasets

Here, we introduce the three datasets used in our experiments in detail. For each dataset, it has publicly released training/validation/test set partitions. Following established practices in previous works, we adopt the same partitions to fairly compare our performance.

- **ScienceQA** [28] is a benchmark for science question answering, which requires machines to reason on a diverse range of science topics. It is collected from elementary and high school science curricula, and contains 21,208 multiple-choice science questions. Most questions are annotated with grounded lectures (83.9%) and detailed explanations (90.5%). The lecture and explanation provide general external knowledge and specific reasons, respectively, for arriving at the correct answer.

- **StrategyQA** [12] is a question-answering benchmark focusing on open-domain questions where the required reasoning steps are implicit in the question and should be inferred using a strategy. StrategyQA includes 2,780 examples, each consisting of a strategy question, its decomposition, and evidence paragraphs. To guide and evaluate the question answering process, each example in StrategyQA was annotated with a decomposition into reasoning steps for answering it, and Wikipedia paragraphs that provide evidence for the answer to each step.

- **GSM8K** [8] is a dataset of 8.5K high quality linguistically diverse grade school math word problems created by human problem writers. The dataset is segmented into 7.5K training problems and 1K test problems. These problems take between 2 and 8 steps to solve, and solutions primarily involve performing a sequence of elementary calculations using basic arithmetic operations to reach the final answer. A bright middle school student should be able to solve every problem. It can be used for multi-step mathematical reasoning.

### B.1.2    Baselines

In this subsection, we introduce the baselines used in our experiments, including Few-shot prompting, Chain-of-Thoughts (CoT) prompting, Tree-of-Thoughts (ToT) and Graph-of-Thoughts (GoT).

- **Few-shot prompting** [5]. Few-shot prompting is a paradigm where the language model is provided with a limited number of examples for a specific task, allowing it to generalize and generate the desired output when presented with new instances of the task. This approach leverages a small amount of task-specific information to guide the language model's behavior and enable it to perform effectively on novel examples.

- **Chain-of-thoughts (CoT)** [45]. CoT prompts the language model to generate intermediate explanations during the reasoning process preceding the final answer. This deliberate emphasis on providing a step-by-step rationale enhances the model's capacity to produce more accurate and contextually grounded results. The inclusion of reasoning explanations contributes to a more robust and insightful generation of answers by guiding the language model through a thoughtful and systematic thinking process.

- **Tree-of-Thoughts (ToT)** [49]. ToT extends the capabilities of language models by enabling deliberate decision-making through the exploration of multiple reasoning paths. It incorporates various search algorithms, allowing the model to traverse diverse routes during the decision-making process. This approach enhances the model's ability to consider alternative perspectives and reasoning strategies, contributing to more nuanced and informed outputs.

- **Graph-of-Thoughts (GoT)** [3]. GoT is an innovative framework that builds upon the advancements introduced by ToT, pushing the boundaries of prompting capabilities in Large Language Models (LLMs). Unlike ToT, GoT represents the information generated by an LLM as an arbitrary graph, introducing a more flexible and comprehensive structure. Furthermore, GoT incorporates an expanded set of thought transformation operations, allowing for a richer and more diverse modeling of the underlying thought processes within the language model.

- **Least-to-most** [56]. The term least-to-most prompting is borrowed from educational psychology, where it is used to denote the technique of using a progressive sequence of prompts to help a student to learn a new skill. The key idea in Least-to-most strategy is to break down a complex problem into a series of simpler subproblems and then solve them in sequence. Solving each subproblem is facilitated by the answers to previously solved subproblems.

- **SelfCheck** [31]. SelfCheck is a zero-shot step-by-step checker for self-identifying errors in LLM reasoning chains. SelfCheck uses the LLM to individually check the conditional correctness of each step in the chain based on the preceding steps, in a manner similar to a human going back to check their working. The results of these individual checks are then integrated to form an overall correctness estimation for the whole reasoning chain.

Table 9: Performance comparison of more baseline methods on MATH dataset.

| Methods (with GPT-4 backbone) | ACCs on MATH datasets |
| --- | --- |
| CoT | 56.99 |
| CoT+SC (sample 5 solutions each time) | 57.24 |
| ToT | 57.18 |
| ToT-variant | 57.02 |
| GoT | 58.78 |
| DeAR | **62.25** |

## B.2 Comparison with More Strong Baselines

In this subsection, we present comparison results with additional strong baselines, including one variant of ToT ("ToT-variant") [27] and CoT with self-consistency ("CoT + SC") [41].We conduct experiments on the more challenging MATH dataset, using GPT-4 as the backbone model, to further demonstrate the effectiveness of our DeAR approach. The MATH dataset is specifically designed to assess the mathematical reasoning and problem-solving abilities of AI models. It consists of 12,500 complex competition-level problems across diverse topics such as algebra, geometry, calculus, number theory, and combinatorics. For our experiments, we use the "gpt-4-0125-preview" version of the GPT-4 model. The results are summarized in Table 9. From the results, we can see that our DeAR method achieves significant improvements over the two newly added baseline methods,

demonstrating the effectiveness of our approach in further enhancing the reasoning capabilities of GPT-4 on more complex reasoning tasks. Moreover, we observe that, based on GPT-4, GoT performs better than both ToT and CoT methods. Although CoT has the lowest ACC, when combined with self-consistency sampling, its performance surpasses that of ToT.

## B.3 Ablation Study of the Self-Check Method in the *Analyze* Stage

Table 10: Performance comparison of DeAR with and without self-check on the ScienceQA dataset.

|  | GPT-3.5 | LLaMA2-7B | ChatGLM3-6B |
|---|---|---|---|
| DeAR w/o self-check | 82.76 | 69.44 | 50.35 |
| DeAR | **83.68** | **70.57** | **51.08** |

The construction of the reasoning tree relies on the indispensable interplay of DeAR's three stages: *Decompose*, *Analyze*, and *Rethink*. An ablation study omitting any of these stages would disrupt the entire process. For instance, removing the *Decompose* stage would prevent both the *Analyze* and *Rethink* stages from functioning, as they depend on analyzing and updating the sub-questions generated during decomposition. Similarly, eliminating the *Analyze* stage would hinder the generation of rationales for each node, rendering the *Rethink* stage infeasible. Without the *Rethink* stage, the first two stages would lose their purpose, reducing the entire framework to a zero-shot approach, where the problem is directly solved at the root node without any further reasoning.

The only reasonable point for ablation is the self-check method within the *Analyze* stage (Section 4.2), as removing it does not structurally affect *Decompose* stage and *Rethink* stage. Therefore, we have included an ablation study on the self-check method using the ScienceQA dataset, as shown in Table 10. The results demonstrate that across various LLM backbones, DeAR consistently outperforms its counterpart without the self-check, confirming the method's effectiveness.

## B.4 Human Evaluation

> **Question:** Janet's ducks lay 16 eggs per day. She eats three for breakfast every morning and bakes muffins for her friends every day with four. She sells the remainder at the farmers' market daily for $2 per fresh duck egg. How much in dollars does she make every day at the farmers' market?
> **Options:**
> (A) Determine the total daily egg production of Janet's ducks, which is 16. Subtract the eggs consumed for breakfast (3) and those used for baking muffins (4). Multiply the remaining eggs by the selling price of $2 per egg to find the daily income at the farmers' market. The answer is $18.
> (B) Janet's ducks lay 16 eggs per day. She uses 3 + 4 = <<3+4=7>>7 eggs per day for breakfast and muffins. She sells 16 - 7 = <<16-7=9>>9 eggs at the farmers' market. She earns $2 for each egg sold, so she makes 9 * $2 = $<<9*2=18>>18 per day. Answer: 18。
> (C) Let's start by calculating the total number of eggs Janet's ducks lay per day: Janet's ducks lay 16 eggs per day. Next, let's calculate how many eggs she uses every morning for breakfast: She eats 3 eggs for breakfast every morning. Now, let's determine how many eggs she uses for baking muffins: She uses 4 eggs to bake muffins for her friends every day. Now, we'll find out the number of eggs that are left to sell at the farmers' market: Total eggs laid - (Eggs eaten for breakfast + Eggs used for baking) = Remaining eggs to sell Remaining eggs to sell = 16 - (3 + 4) Remaining eggs to sell = 16 - 7 Remaining eggs to sell = 9. Now, we know she sells each egg for $2: Total money made = Number of remaining eggs * Price per egg. Total money made = 9 * $2. Total money made = $18. Answer: 18.
> **Please select the one that has the best logical coherence.**

Figure 6: The interface of our Human Evaluation method.

In Section 5.4, we analyze the logical coherence of rationales generated by our framework. We randomly sampled 100 questions from each of the three datasets, and for each question, we used ToT, GoT, and DeAR to generate three distinct rationales. Each question, along with the rationales from ToT, GoT, and DeAR, was presented to 10 well-educated annotators (all with at least a bachelor's degree). To ensure impartiality, we did not disclose the model that generated each rationale, and the order of the rationales was randomized. The annotators were asked to select the rationale they found most logical, following a majority-vote approach: each question was annotated by all 10 annotators, with the rationale receiving the highest vote count chosen as the final result. Figure 6 shows the template used for annotation. The annotation achieved a Kappa score of 0.70, indicating good agreement among annotators.

Table 11: Comparison of Avg API calls and ACC between DeAR, ToT, and GoT on ScienceQA.

|  | DeAR | ToT (b=2, d=4) | GoT (b=2, d=4) |
|---|---|---|---|
| **Avg API calls** | **9.82** | 11.35 | 13.74 |
| **ACC** | **0.837** | 0.826 | 0.831 |

## B.5 More Discussion about Efficiency

In Section 5.7, we compared the reasoning time and accuracy of DeAR with ToT and GoT across different datasets based on ChatGLM3-6B, concluding that DeAR achieves a better trade-off between time and accuracy. To further validate this conclusion, we measure the average number of API calls made by DeAR, ToT, and GoT for each question in ScienceQA based on GPT-3.5, along with their reasoning ACCs, as shown in Table 11. It's clear that DeAR makes fewer API calls on average, which implies less time under the same conditions, while simultaneously achieving higher ACCs.

## B.6 Case Study

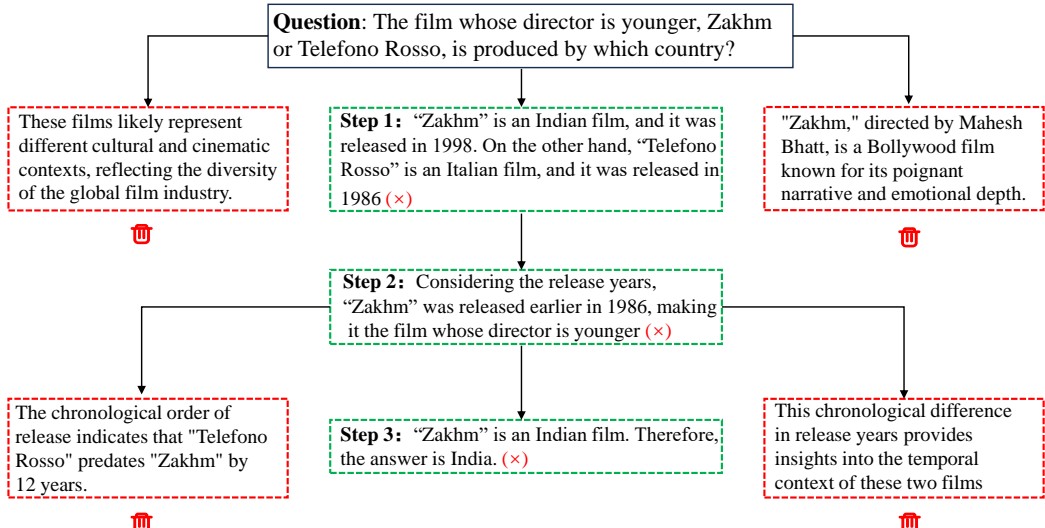

Figure 7: Case of ToT (GPT-3.5)'s reasoning process.

We conduct case study to demonstrate that our framework's reasoning process aligns more with human logic, effectively reduces logical errors, and is more interpretable. We present one typical intricate reasoning case from StrategyQA with GPT-3.5 backbone in Figure 7 and Figure 8. As shown in Figure 7, ToT generates a fixed number of thoughts at each level (3 in this case), and it stops generating at a depth of 3. It searches for an optimal path ("Step 1", "Step 2" and "Step 3") from the tree structure as the reasoning process and discards other thoughts. However, in this example, the logical relationships between the generated thoughts are not clear (e.g., "Question" and "Step1") and there are logical errors in intermediate step "Step 1" leading to errors in the subsequent step ("Step 2" and "Step 3").

In contrast, DeAR establishes a clear logical structure through *Decompose* stage (black directed arrows). For example, to answer the comparison question "#2": Which of these two directors has a smaller age?", DeAR decomposes it into more fine-grained sub-questions "#3": What is the age of Zakhm's director?" and "#4": What is the age of Telefono Rosso's director?". Then, with *Analyze* stage, we obtain the rationales (texts in the green dashed box) for each sub-question, which are then utilized in the *Rethink* stage to update the existing upper-level rationales (green dashed arrows). For example, the rationale "Nanni Moretti directed Telefono Rosso." for sub-question "#6" is used to update the rationales of "#2","#1" and the original "Question". The three-stage cycle iterates until no questions can be further decomposed and we get the tree structure as shown in Figure 8. Finally, we

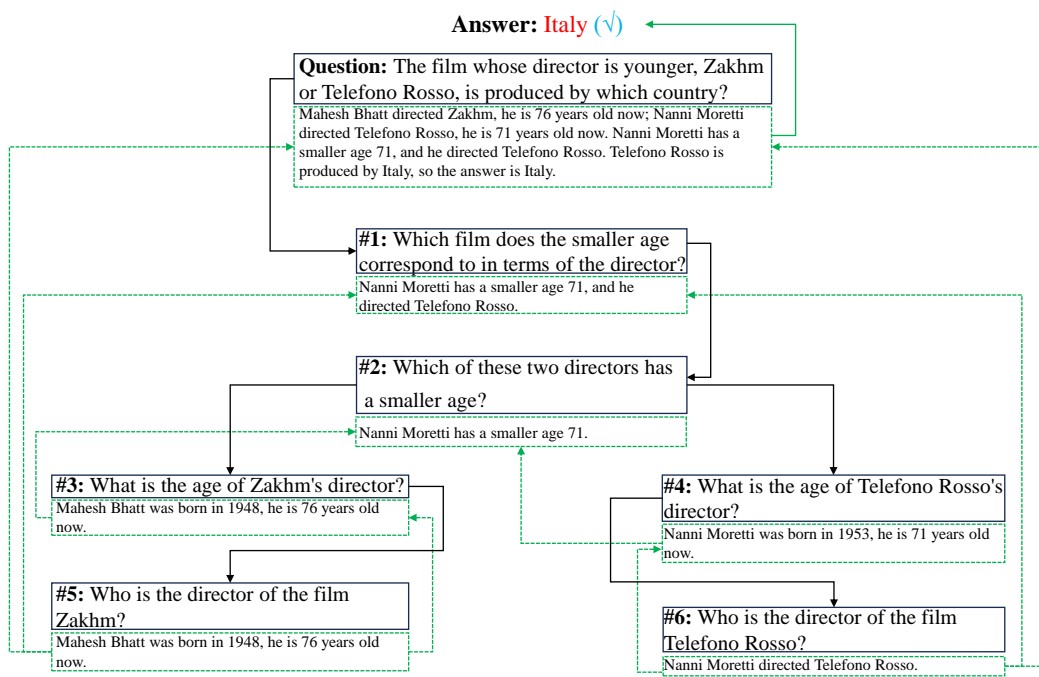

Figure 8: Case of DeAR (GPT3.5)'s reasoning process.

can extract the answer "Italy" (correct in this case) from the rationale of the original question at the root node. Our reasoning structure effectively avoids the errors generated by ToT, while being more logical and interpretable.

We further demonstrate the error correction process in the reasoning during the *Rethink* stage for previously generated nodes, as illustrated in Figure 9. The left part shows the state of the reasoning tree when solving sub-question "#2". At this point, the original question and the answers to "#1" and "#2" are incorrect (marked in red). The right part shows the state after decomposing "#2" into sub-questions "#3" and "#4", solving them, and updating the reasoning tree. The purple text represents the rationales obtained from solving the newly decomposed sub-questions "#3" and "#4", which are subsequently used to update the rationale for "#2", correcting the wrong answer "Mahesh Bhatt has a smaller age 70" to "Nanni Moretti has a smaller age 71". This correction also impacts earlier nodes, replacing the original wrong answer with the correct one (with the correct parts shown in blue). Without using the newly obtained rationale to update the previous nodes in DeAR, the above errors would not have been corrected. Therefore, the *Rethink* stage is crucial for DeAR to achieve accurate results in reasoning.

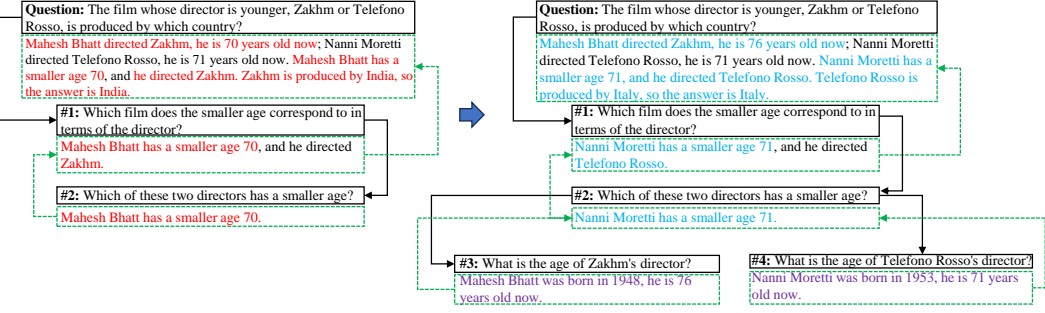

Figure 9: Case of DeAR (GPT3.5)'s error correction process in the *Rethink* stage.

## C   Broader Impacts

This work endeavors to advance the field of natural language processing through the introduction of a novel DeAR (*Decompose-Analyze-Rethink*) reasoning framework, leveraging LLMs for enhanced reasoning capabilities. The potential broader impact of our research lies in its implications for natural language understanding and reasoning systems. By dynamically generating and updating rationales, our framework contributes to the development of more effective and interpretable language models.

The societal consequences of our work include the potential improvement in the interpretability and reliability of machine-generated reasoning, which can have positive implications across various domains, such as education, decision support systems, and natural language processing applications. However, it is essential to approach these advancements with a critical lens, considering the ethical implications and societal impact of widespread deployment.

## D   Limitations and Future Work

While the DeAR framework can significantly enhance reasoning capabilities for large language models, several limitations merit attention. First, as shown in Figure 5, although DeAR achieves a better trade-off between time and accuracy, the reasoning time for complex problems remains relatively long. This is due to the significant overhead associated with the iterative cycles of the *Decompose*, *Analyze*, and *Rethink* stages. Therefore, there is still room for improvement in enhancing reasoning efficiency for practical applications in the future. Second, in the *Decompose* Stage, while logic heuristics contribute to overall performance improvement (as illustrated in Table 6), constructing these heuristics requires additional annotation. For the datasets used in this paper, most of the training data already includes annotated question decompositions, saving considerable time in preparation. However, extending DeAR to other datasets may necessitate the development of more efficient methods to reduce or eliminate the need for the annotation. Third, while DeAR outperforms existing methods like ToT and GoT in flexibility and error reduction, its real-world applicability requires further validation across a broader range of datasets to fully assess the framework's versatility and robustness, as well as to explore the potential of more LLMs for complex reasoning. In the future, we may consider exploring methods to enhance LLMs' reasoning capabilities on more complex tasks, such as those in STEM fields and programming. Recent studies, including OpenAI's o1 model, utilize large-scale reinforcement learning algorithms to teach the model how to think productively through its chain of thought. Based on these advancements, a promising approach would be to investigate how to integrate similar methods with the model's self-thinking capabilities to enhance adaptive learning and facilitate continuous improvement in reasoning.

