# OpenReview forum: "Decompose, Analyze and Rethink: Solving Intricate Problems with Human-like Reasoning Cycle"
_NeurIPS.cc/2024/Conference — NeurIPS 2024 oral_

### Official Review · Reviewer_ANC9 · 2024-06-28

**Soundness:** 4
**Presentation:** 3
**Contribution:** 3
**Rating:** 7
**Confidence:** 4

**Summary:**

The paper introduces a reasoning framework called Decompose-Analyze-Rethink (DeAR) for enhancing the reasoning capabilities of large language models (LLMs). DeAR mimics human cognitive reasoning by decomposing complex problems into simpler sub-problems using a Reasoning Tree structure, analyzing these sub-problems independently, and rethinking the answers in light of new insights from sub-problem solutions. This iterative cycle allows for dynamic adjustments and error corrections in the reasoning process. The proposed framework shows promising results across multiple reasoning datasets.

**Strengths:**

1) The paper makes solid progress on improving how LLMs solve complex problems, an important area in AI research.
2) The paper is well-organzied and the proposed method is well-explained. The entire idea is reasonable and well-aligned with human thinking.
3) The experiments are sound. The proposed method is thoroughly tested on different types of complex problems and the performance improvement over SOTA methods (e.g., CoT, ToT and GoT) is significant.

**Weaknesses:**

1) The process for obtaining decomposition demonstrations in the logic heuristics lacks a detailed explanation.
2) As mentioned in Section 2.2, there are also other works that explore problem decomposition in LLMs. Lacking further discussion or evaluations on how their work differs from existing paradigms of problem decomposition may limit the technical contribution of the paper.
3) The experimental design could be enhanced by providing a more detailed analysis. For example, the effectiveness of the “self-check” mechanism is not well evaluated. The authors may consider showing the error rates of the generated rationales and demonstrate how the “self-check” stage contributes to reducing these errors.
4) The paper could benefit from improvements in presentation. For example: a) Table 4 is incorrectly referenced within the text, b) a typo in Algorithm 1 (stgt;).

**Questions:**

Please address the concerns outlined in Weaknesses.

**Limitations:**

No negative societal impact

---

> ### Author Rebuttal · Authors · 2024-08-06
>
> We appreciate your affirmation of the contribution of our paper. For your concerns:
>
> **Q1**: The process for obtaining decomposition demonstrations in the logic heuristics lacks a detailed explanation.
>
> **A1**: Thank you for your comments. We have provided a detailed explanation for the obtaining of question decomposition demonstrations and examples of prompts for the Decomposition Stage in **Appendix B.1**. We use a BERT encoder to transform target question Q and human-annotated question decomposition demonstrations into vector representations. Then we use cosine similarity to select the top K (K=3 in our setting) similar demonstrations as logic heuristics.
>
> **Q2**:  Lacking further discussion or evaluations on question decomposition.
>
> **A2**: Thank you for your insightful comments. Other methods that use question decomposition to solve complex problems typically decompose the original question into sub-questions through a simple prompting method and then solve them step by step, such as the least-to-most approach. In contrast, the DeAR framework we propose not only uses logic heuristics during the **Decompose** stage to enhance the logic of the question decomposition but also provides more refined planning and updating of the problem-solving process during the **Analyze** and **Rethink** stages. This further ensures the reliability of the problem-solving process, preventing the spread of errors. Additionally, the reasoning tree generated by DeAR makes the reasoning process more interpretable.
>
> **Q3**:  The effectiveness of the “self-check” mechanism is not well evaluated.
>
> **A3**: Thank you for your insightful comments. Here, we verify the effectiveness of the self-check method by comparing the predict accuracy of DeAR and DeAR(w/o self-check) on the ScienceQA dataset. DeAR(w/o self-check) refers to the version where the self-check part is removed during the Analyze stage, while the rest of the implementation process is identical. The experimental results are shown in the following table, from which it can be seen that under different backbones, DeAR has higher predict accuracy, demonstrating the necessity of the self-check method for correcting errors.
>
> |     |     | ScienceQA|      |
> |  ----  | ---- | ---- |  ----  |
> |     | GPT3.5|LLaMA2-7B|ChatGLM3-6B|
> |**DeAR w/o self-check**| 82.76 | 69.44  | 50.35 |
> |**DeAR**| 83.68  | 70.57  | 51.08  |
>
> **Q4**: The paper could benefit from improvements in presentation.
>
> **A4**: Thank you very much for your suggestions. We will carefully review and correct any writing errors.

---

> ### Author Response · Authors · 2024-08-13
> **Thank You for Any Valuable Feedback**
>
> Thank you for your insightful feedback once again. We hope that our response addresses your concerns and questions. As our author-reviewer discussion nears its end, we'd appreciate knowing if your concerns are resolved. We are open for any further discussion if needed.

---

> > ### Comment · Reviewer_ANC9 · 2024-08-13
> >
> > Thank you for your response. I do not have further concerns and I will keep my positive score.

---

### Official Review · Reviewer_Zsa1 · 2024-07-11

**Soundness:** 3
**Presentation:** 3
**Contribution:** 3
**Rating:** 7
**Confidence:** 3

**Summary:**

The paper proposes DeAR-prompting (decompose, analyze, rethink) as a new prompting paradigm. The framework basically consists of decomposing the original question into subquestions, answering the subquestions and analyzing the answers and potentially rethinking answers to earlier questions based on the new answers to correct mistakes. The approach is evaluated experimentally and shows significant improvement over ToT and GoT prompting on three benchmarks.

**Strengths:**

I find the idea very intuitive, well motivated and mostly well presented. The experiments indicate that DeAR improves the state-of-the-art and this seems intuitively plausible.

**Weaknesses:**

From my perspective, there are some weaknesses, but most of them I wouldn't consider as serious issues, but rather as starting points for future work.
- one problem is decomposing the question. This step can probably itself be improved by different prompting strategies. In B1, I was indeed surprised by the first decomposition example. It seems to me that there is nothing that suggests that the Pantheon is a Mausoleum or that it is reserved for citicens of a particular country. It seems to me that the example is much more than just a decomposition as it involves a lot of Background Knowledge that may or may not be available. The second and third example seem much more natural.
- another problem is evaluating the answers. It seems very naive to ask the LLM for a score. While some People claim that LLMs can give meaningful quantitative evaluations, there is also a lot of evidence to the contrary. Given the nature of LLMs, it seems to me that there is a good chance that the LLM will just return a score that frequently occured in "similar contexts" during training and is not particularly meaningful. The idea of applying voting methods for this step sounds more convincing to me.
- the paper currently argues that particular stages are important because  they do not exist in other frameworks (e.g., "the improvements over ToT highlight the advantage of Decompose stage"). It would be more convincing to do an ablation study.
- there are quite a few typos and grammatical problems in the paper. It would be good to apply a spell checker. Just two examples:
 ** line 114: an novel -> a novel
 ** line 257: Graph-of-Thoughtss -> Graph-of-Thoughts?
- finally, a minor philosophical point that will not affect my evaluation: personally, I am not a fan of the whole ANN-vs-human discussion. A neuron in an ANN is just a numerical parameter, a human neuron is a biological cell, which can itself be seen as a primitive life form with its own metabolism. Perhaps something intelligent will evolve from ANNs, but comparing them to biological NNs seems rather far-fetched to me. I appreciate that the paper does not really go into that direction, but does the whole human cognitive reasoning discussion really add anything to the paper? I agree that the proposed approach is more natural than other prompting approaches, but does it really resemble what humans do? Decomposition is certainly a part of what humans do, but do we really have to go back and revise our previous answers because we hallucinated a random answer at some point? This does not really seem to be a reasoning problem in general, but an artifact of the probabilistic-generative nature of LLMs. Of course, it's important to deal with this in LLMs, but do we really need to sell this as human-like?

**Questions:**

I was surprised by the first question example in B1. Was there a rationale for adding so much information in the decomposition example that goes beyond the original question? It's hard to evaluate how often this really happens experimentally, but did you look into some examples to see how far the subquestions go beyond the original question?

**Limitations:**

Yes

---

> ### Author Rebuttal · Authors · 2024-08-07
>
> We appreciate your affirmation of the motivation of our idea, and the implementation of our approach DeAR. As for your concerns:
>
> **Q1**: Question decomposition examples.
>
> **A1**: Thanks for your insightful comments. In our approach, the logic heuristics provided in the problem decomposition prompt vary dynamically depending on the question. The logic heuristics presented in Table 5 are specific to a particular case of a question. For a given question, we select problem decomposition examples from the demonstration pool based on the calculated cosine similarity. Different examples may be selected as part of the prompt for different questions (see Appendix B.1 for a detailed description). This method can effectively adjust the prompt according to the differences in questions, allowing for better decomposition tailored to the characteristics of each problem compared to a fixed prompt.
>
> **Q2**: Using LLMs for answer scoring.
>
> **A2**: Thanks for your valuable comments. In section 4.2, line 217, we mentioned that scoring for the answer could also be achieved using other voting or classification methods. In the ToT method, a comparison was made between "value" and "vote" scoring approaches, demonstrating that both are effective for verifying the accuracy of answers[1]. In our paper, for simplicity, we explored the method of directly generating scores using the backbone LLMs , and we will supplement the results obtained using voting in the future.
>
> [1] Tree of thoughts: Deliberate problem solving with large language models.
>
> **Q3**: An ablation study about Decompose Stage.
>
> **A3**: Thanks for your insightful comments. Given that each stage in our method is essential for constructing the reasoning tree, it is challenging to perform an ablation study by simply removing one stage. If we were to conduct an ablation study that eliminates the decompose stage, then the subsequent analyze and rethink stages would be hindered, because these stages rely on analyzing and updating the sub-questions that result from the decomposition process. A possible way to validate the effectiveness of the decompose stage is to replace its prompt with prompts from other methods, such as those used for problem decomposition in the Least-to-Most approach. In the table below, we have included supplementary comparative experiments on ScienceQA that demonstrate the superiority of our designed decompose stage. As shown in the table, the performance will decline after we replace our prompt with Least-to-Most decomposition prompt, indicating the effectiveness of our method. We will consider designing additional experiments to further validate the effectiveness of different stages.
>
> |  | DeAR+GPT3.5| DeAR+GPT3.5 (Least-to-Most decomposition prompt)|
> |  ----  | ---- | ---- |
> |Accs on ScienceQA |83.68 | 81.33 |
>
> **Q4**: Typos and grammatical problems.
>
> **A4**: Thanks for your suggestions, we will correct these errors in the revised version.
>
> **Q5**: Human cognitive reasoning discussion.
>
> **A5**: Thank you for your insightful comments and for raising a philosophical consideration regarding the comparison between ANNs and human cognition. Regarding the human cognitive reasoning discussion in our paper, we included it with the intention of drawing parallels to human problem-solving strategies, like the decompose, analyze and rethink stages, which can provide intuitive understanding and potentially guide the development of more natural and effective AI systems.
>
> **Q6**: The first question example in B1.
>
> **A6**: Thank you for your comments. For each question, we employ cosine similarity to select the most semantically similar questions from the demonstration pool to construct decomposition examples. For instance, regarding the original question "Does the actress who played Elizabeth II speak fluent Arabic?", the questions chosen from the demonstration pool are "Will Queen Elizabeth be buried in the Pantheon?", "Was Elizabeth II the Queen during the Persian Gulf War?", and "Does Elizabeth II reign over the Balearic Islands?". These selected questions and their decomposition examples might contain additional information. However, compared to direct prompting methods (such as using the least-to-most decomposition prompt for prompting), this method is more effective, as we have also demonstrated in our response table for Q3.

---

> ### Author Response · Authors · 2024-08-13
> **Thank You for Any Valuable Feedback**
>
> Thank you for your valuable feedback. We hope that our response has effectively addressed your concerns and questions. As our author-reviewer discussion comes to a close, we would appreciate if your concerns have been resolved. We are always open to further discussion if necessary.

---

> > ### Comment · Reviewer_Zsa1 · 2024-08-13
> >
> > Thank you for the clarifications. As I wrote in my review, I do not have any serious concerns about this paper and remain on the acceptance side.

---

### Official Review · Reviewer_WGVX · 2024-07-12

**Soundness:** 3
**Presentation:** 4
**Contribution:** 3
**Rating:** 7
**Confidence:** 5

**Summary:**

This paper proposes a recursive method for LLMs to solve complex reasoning tasks. The approach formulates problem-solving as a hierarchical tree structure, where each problem is broken down into a tree of sub-problems. Each sub-problem is then analyzed and updated. This method has been evaluated on datasets such as ScienceQA, StrategyQA, and GSM8K, demonstrating improved accuracy on LLMs like Llama-2, GPT-3.5, and ChatGLM3.

**Strengths:**

S1. The concept of the proposed framework is sound, and the cycle algorithm is shown with clear examples. The main idea, which mimics human reasoning, is easy to understand and is presented in a straightforward way.

S2. The performance improvements over SOTA methods like ToT and GoT are significant. The experiments on different LLMs (GPT3.5, Llama2, ChatGLM3) also demonstrate the method’s versatility.

S3. The structure of the framework is more flexible and reasonable compared to CoT, ToT and GoT. The method can generate the reasoning path based on the specific logic of the problems and timely correct errors.

**Weaknesses:**

W1. The effectiveness of the “self-check” method in “Analyze Stage” may need further validation. The paper (Jie Huang et al, "Large Language Models Cannot Self-Correct Reasoning Yet") show that LLMs cannot correct themselves.

W2. Could the authors provide a more detailed explanation of each step of the algorithm's execution, including the input and output results, in the case study? For instance, in the example in Figure 9, only the final reasoning process of each node is shown. It would be better if the authors could explain how the contents of these nodes are updated.

**Questions:**

Q1. Is the method effective for more complex tasks? In the context of math reasoning, the community might be more interested in results on MATH or MathQA, as opposed to GSM8K, which is relatively simple for models like GPT-3.5.

Q2. The paper provides an efficiency analysis based on ChatGLM3. Could the authors provide a more detailed analysis based on GPT-3.5? For example, could they present a comparison of the number of API calls and the number of input tokens compared to ToT and GoT? This is beneficial to verify the method's efficiency on API-based LLMs.

**Limitations:**

The authors addressed the limitations in Appendix D: 1. The self-check method may add more computational complexity. 2. The autonomy in generating branches might result in inconsistency in the reasoning quality. 3. A broader range of datasets should be considered to validate its real-world applicability.

---

> ### Author Rebuttal · Authors · 2024-08-06
>
> We appreciate your acknowledgement of on our study motivation, model design, experimental results, and presentation. Your suggestions are insightful for us.
>
> **Q1**:  The effectiveness of the “self-check” method in “Analyze Stage” may need further validation.
>
> **A1**: Thank you for your insightful comments. Here, we add an ablation study focusing on the self-check mechanism, utilizing the ScienceQA dataset for our analysis, as illustrated in the table below. The results demonstrate that across three LLM backbones, the DeAR model outperforms its counterpart without the self-check method, thereby validating the self-check method's efficacy. Although other methods that employ LLMs for self-correction may not be sufficiently effective, our experiments have demonstrated that incorporating a self-check method during the Analyze stage is very necessary. We intend to incorporate these findings into the updated version of our work.
>
> |     |     | ScienceQA|      |
> |  ----  | ---- | ---- |  ----  |
> |     | GPT3.5|LLaMA2-7B|ChatGLM3-6B|
> |**DeAR w/o self-check**| 82.76 | 69.44  | 50.35 |
> |**DeAR**| 83.68  | 70.57  | 51.08  |
>
> **Q2**: More detailed explanation of case studies.
>
> **A2**: Yes. We'll break down how DeAR enhances the reasoning process with a real example from Figure 9 in Appendix C.4. Imagine we need to solve a comparison question, "#2": "Who is younger between these two directors?" In the **Decompose** stage, DeAR breaks "#2" into simpler sub-questions, "#3" and "#4", asking for the ages of the directors of "Zakhm" and "Telefono Rosso," which are more manageable for Large Language Models (LLMs) to figure out.
> Once we have the answers to these sub-questions, DeAR moves on to the **Analyze** stage. Here, it not only gets the specifics but also spots and fixes a mistake: it corrects the age of Mahesh Bhatt from 70, born in 1954, to the accurate 76 years old, born in 1948. With the correct information in hand, in **Rethink stage**, DeAR then revisits the original question and makes the necessary update, correcting the initial guess of "Mahesh Bhatt" to the right answer, "Nanni Moretti."
> This step-by-step approach allows DeAR to catch and correct any faulty reasoning along the way, stopping errors from spreading. In the next version of our work, we'll add more such detailed examples to paint a clearer picture.
>
> **Q3**: Is the method effective for more complex tasks?
>
> **A3**: Thank you for your comments. We conduct further experiments based on GPT-4, particularly on the more challenging MATH dataset, to address your inquiry. The results are presented in the table below. For more complicated questions in MATH, DeAR also performs better.
>
> | Methods (with GPT-4 backbone) | ACCs on MATH |
> |  ----  | ---- |
> |CoT     | 56.99  |
> |CoT+SC **[1]** (sample 5 solutions each time)  | 57.24  |
> |ToT     | 57.18   |
> |ToT-variant **[2]** | 57.02  |
> |GoT     | 58.78  |
> |DeAR    | **62.25**  |
>
> **[1]** Self-consistency improves chain of thought reasoning in language models.
>
> **[2]** Large language model guided tree-of-thought.
>
> **Q4**: Could the authors provide a more detailed efficiency analysis based on GPT-3.5?
>
> **A4**: Yes. On the ScienceQA dataset, using GPT3.5 as backbone, to ensure a fair comparison, we compare DeAR (with parameters b=1.58 and d=3.62) with ToT, which has the closest values for branch b and depth d (b=3, d=4). We've looked at the average number of API calls for each question, and ACCs on test set, as shown in the table below. It's clear that our method makes fewer API calls on average, which means less time under the same conditions, and achieves higher ACCs at the same time. We'll add more detailed experiments about the average input tokens in the updated version.
>
> |  | DeAR|  ToT (b=2, d=4) | GoT (b=2, d=4) |
> |  ----  | ---- | ---- | ---- |
> |**Avg API calls**  | **9.82** | 11.35 | 13.74
> |**ACC**  | **0.837** | 0.826 | 0.831

---

> ### Author Response · Authors · 2024-08-13
> **Thank You for Any Valuable Feedback**
>
> Thank you for your constructive feedback. We sincerely hope our response has answered your concerns and questions. As we near the end of this discussion, we would appreciate it if you could let us know whether all your concerns have been addressed. We are open to further discussion if needed.

---

> > ### Comment · Reviewer_WGVX · 2024-08-14
> >
> > Thanks for the detailed rebuttal. I will keep my positive opinion on this paper.

---

### Official Review · Reviewer_vfU8 · 2024-07-12

**Soundness:** 3
**Presentation:** 3
**Contribution:** 2
**Rating:** 6
**Confidence:** 4

**Summary:**

The paper presents **DeAR**, a new reasoning framework for large language models to perform intricate reasoning tasks. Inspired by human cognition, it decomposes problems into sub-questions within a Reasoning Tree, refining solutions through iterative Decompose-Analyze-Rethink cycles. Compared to existing state-of-the-art approaches like **ToT** and **GoT**, **DeAR** offers more flexibility and continuous rationale refinement, leading to reduced logical errors and improved performance across various reasoning benchmarks.

**Strengths:**

- A novel reasoning framework implemented with a **Decompose-Analyze-Rethink** (**DeAR**) cycle has been proposed to enhance the capabilities of LLMs in solving intricate problems.
- The proposed framework is capable of generating rationales with better logical consistency while achieving better accuracy in less time per question.
- Extensive experiments on three complex reasoning benchmarks demonstrate the superiority of **DeAR** over state-of-the-art approaches (e.g., **ToT**, **GoT**), showcasing its ability to improve performance for intricate reasoning with different LLMs.

**Weaknesses:**

- Lack of ablation studies to analyze the contribution of each individual step, i.e., ***Decompose***, ***Self-Check*** and ***Rethink***.
- A small number of participants in human evaluations leads to statistically unreliable conclusions. I notice that different prompting methods elicit the LLM to produce responses of different lengths, which is also a confounding factor that can affect the choice, as humans prefer more concise responses.
- The values for threshold hyperparameters, i.e., $\epsilon\_1$ and $\epsilon\_2$ should be carefully set.
- Typo. In Line 322, *Table 3* should be *Tabel 4*.

**Questions:**

- How large is the (human-annotated question decomposition) demonstration pool for each of the datasets?
- I notice that the authors employ a cosine similarity-based strategy to pick appropriate demonstrations when constructing prompts at the ***Decompose*** stage, is the same strategy used to test the performance of the baseline prompting methods?
- How to assign the proper values for $\epsilon\_1$ and $\epsilon\_2$ if there is no validation set available?

**Limitations:**

Yes, the authors have covered several limitations in Appendix 4.

---

> ### Author Rebuttal · Authors · 2024-08-06
>
> We appreciate your positive comments on the novelty and efficiency of our DeAR and the affirmation of its superior performance over SOTA methods.
>
> **Q1**: Lack of ablation studies to analyze the contribution of each individual step.
>
> **A1**: Thanks for your insightful comments. The construction process of our proposed reasoning tree is such that the three stages—decompose, analyze, and rethink—are indispensable. If we conduct a ablation study that omits one of these stages, for example, removing the decompose stage, then both the analyze stage and the rethink stage would be unable to proceed, as the latter two stages must analyze and update the sub-problems generated from the decomposition. Similarly, eliminating the analyze stage would result in the inability to obtain the rationales for each node, thereby preventing the rethink stage from taking place. Removing the rethink stage would also render the first two stages pointless; the entire framework would then devolve into using a zero-shot approach to directly solve the problem at the root node and obtain the result.
>
> Here, the only part where an ablation study can be reasonably conducted is the self-check method within the analyze stage, as removing self-check will not structurally affect the other two stages. Therefore, we have added an ablation study on the self-check method using the ScienceQA dataset, as shown in the table below. It can be observed that, based on different LLM backbones, DeAR consistently performs better than DeAR without self-check, which also proves the effectiveness of the self-check method. We will include this experiment in the updated version.
>
> |     |     | ScienceQA|      |
> |  ----  | ---- | ---- |  ----  |
> |     | GPT3.5|LLaMA2-7B|ChatGLM3-6B|
> |**DeAR w/o self-check**| 82.76 | 69.44  | 50.35 |
> |**DeAR**| 83.68  | 70.57  | 51.08  |
>
> **Q2**: Human evaluations.
>
> **A2**: Thank you for your comments. We have adopted some approaches to help minimize the bias of annotators towards rationales of varying lengths. For example, we provided each annotator with detailed annotation instructions, allowing them to select the most logical response from the answers given by different models, as shown in Figure 6, Appendix C.2. At the same time, we performed multiple random samplings from the dataset, each time with a different set of annotators, to further prevent the unreliability of results due to the subjective factors of individual annotators. We have five annotators, which is a similar number to that used in other studies employing human evaluation methods, such as in [1]. We will include more details about the sampling and annotation process in the updated version.
>
> [1] Guiding Mathematical Reasoning via Mastering Commonsense Formula Knowledge
>
>
> **Q3**: The values for threshold hyperparameters.
>
> **A3**: Thank you for your insightful comments. We set the thresholds by conducting the threshold combination experiment in Section 5.6. We selected the threshold combination that yields the highest reasoning accuracy for our configuration.
>
> **Q4**: Typo about table’s name: Table 3 should be Tabel 4.
>
> **A4**: Thank you for the reminder, we will correct it in the updated version.
>
> **Q5**: How large is the demonstration pool?
>
> **A5**: For the ScienceQA dataset, we randomly selected some questions from each topic in the training set and annotated 500 examples as a demonstration pool. For GSM8K and StrategyQA, since their training sets already have annotations for problem decomposition, we directly chose 500 items from them as the demonstration pool. We will include this in the updated version.
>
> **Q6**: I notice that the authors employ a cosine similarity-based strategy to pick appropriate demonstrations when constructing prompts at the Decompose stage, is the same strategy used to test the performance of the baseline prompting methods?
>
> **A6**: Thank you for your question.  Selecting demonstrations for problem decomposition prompts will only be effective for methods that include a problem decomposition step. Among the baselines in this experiment, only the least-to-most method includes a problem decomposition step. Therefore, for least-to-most, we use the same demonstration pool and cosine similarity selection method as DeAR. As for other baselines, such as CoT, ToT, GoT, since they do not include a problem decomposition step, naturally we do not select decomposition demonstrations for prompting.
>
> **Q7**: How to assign the proper values if there is no validation set available?
>
> **A7**: Thank you for your question. A portion of the data from the training set can be selected as a validation set to verify the effects of different threshold combinations, and the optimal combination can be chosen for testing on the test set.

---

> ### Author Response · Authors · 2024-08-13
> **Thank You for Any Valuable Feedback**
>
> We greatly appreciate your feedback and hope that our responses have addressed your concerns. As we approach the end of our author-reviewer discussion, we would be grateful if your concerns have been resolved. We remain available for any further discussion if needed.

---

> > ### Comment · Reviewer_vfU8 · 2024-08-14
> > **Thank you for the rebuttal**
> >
> > Thanks for the detailed clarifications. After reading all reviews and rebuttals I found that all my concerns have been well resolved. I would like to keep my rating.

---

### Official Review · Reviewer_dFYv · 2024-07-13

**Soundness:** 3
**Presentation:** 3
**Contribution:** 3
**Rating:** 7
**Confidence:** 4

**Summary:**

The paper presents a novel reasoning framework DeAR (Decompose-Analyze-Rethink), which aims to advance the capabilities of large language models (LLMs) in handling complex reasoning tasks. DeAR introduces a Decompose-Analyze-Rethink cycle that involves breaking down intricate problems into simpler sub-questions, analyzing these to form rationales, and revisiting prior answers to refine the reasoning process. Different from the rigid structures of existing methods like Tree-of-Thoughts (ToT) and Graph-of-Thoughts (GoT), this approach allows each branch to be independently generated without preset configurations, thereby enhancing logical coherence. Extensive experimentation on several benchmarks are conducted to demonstrate the effectiveness of the framework.

**Strengths:**

1.The DeAR framework introduces a novel reasoning cycle that mimics human cognitive reasoning, offering a fresh perspective on how LLMs can tackle complex problems.

2.By decomposing problems into sub-questions and rethinking rationales, DeAR ensures greater logical consistency compared to traditional methods like ToT and GoT.

3.Experimental results show that DeAR achieves significant improvements over state-of-the-art methods, particularly in reducing logical errors and enhancing the reasoning process with different LLMs.

4.By constructing a reasoning tree through a three-stage framework, DeAR provides a clear and interpretable reasoning process, which aids in understanding the decision-making of LLMs.

**Weaknesses:**

1. While the framework is designed to enhance reasoning accuracy and flexibility, the iterative nature of the cycle may lead to increased computational demands, particularly when dealing with highly complex problems. The authors can further discuss this point.

2. The paper should include experimental comparisons between the DeAR framework and more strong baselines, such as other variants of ToT and CoT+SC approach, an enhanced Chain-of-Thoughts approach that incorporates self-consistency checks [1][2].
[1] Long J. Large language model guided tree-of-thought[J]. arXiv preprint arXiv:2305.08291, 2023.
[2] Mo S, Xin M. Tree of uncertain thoughts reasoning for large language models[C]//ICASSP 2024-2024 IEEE International Conference on Acoustics, Speech and Signal Processing (ICASSP). IEEE, 2024: 12742-12746.

**Questions:**

1. Can the authors present more case studies where the DeAR framework has been or could be effectively applied, and how the problem-solving process might benefit from the enhanced reasoning capabilities?

2. Does the method work with stronger LLMs? The paper should present the framework performance with GPT4 as its backbone to validate its effectiveness. If the computation cost is too high, the authors can consider running experiments on subsets of the original datasets.

**Limitations:**

The authors have discussed the limitations of the proposed DeAR framework, including potential computational overhead, variability in reasoning quality, and the need for broader real-world testing.

---

> ### Author Rebuttal · Authors · 2024-08-06
>
> We appreciate your affirmation of the motivation of our paper, the significance of our experimental results and the novelty of DeAR.
>
> **Q1**: The iterative nature of the cycle may lead to increased computational demands.
>
> **A1**: Thank you for your comments. In section 5.7, we compare the efficiency of our framework with different variants of ToT/GoT with ChatGLM3-6B backbone. As shown in Figure 5, compared to state-of-the-art ToT/GoT methods, the point corresponding to our method achieve better ACC with less time.
>
> **Q2**: Include comparisons with more strong baselines (one variant of ToT**[1]** and CoT+SC**[2]**).
>
> **A2**: Thank you for your comments.  We conducted further comparisons with one variant of ToT**[1]** and CoT+SC**[2]** based on GPT-4 backbone, on a more challenging dataset MATH, to address your inquiry. The results are presented in the table below.
>
> | Methods (with GPT-4 backbone) | ACCs on MATH |
> |  ----  | ---- |
> |CoT     | 56.99  |
> |CoT+SC **[1]** (sample 5 solutions each time)  | 57.24  |
> |ToT     | 57.18   |
> |ToT-variant **[2]** | 57.02  |
> |GoT     | 58.78  |
> |DeAR    | **62.25**  |
>
> **Q3**: More case studies.
>
> **A3**: Here, we use the case in Figure 9, Appendix C.4, to further explain how the reasoning process benefit from DeAR’s decomposition, analyze and rethink. First, to answer the comparison question “#2”: Which of these two directors has a smaller age?”, our framework decomposes it into sub-questions “#3”: What is the age of Zakhm’s director?” and “#4”: What is the age of Telefono Rosso’s director?”, which are easier for LLMs to solve. Second, in the Analyze stage, DeAR obtains the answers of sub-question #3 and #4, and also corrects the wrong answer “Mahesh Bhatt was born in 1954, he is 70 years old now” to the right one “Mahesh Bhatt was born in 1948, he is 76 years old now”. After that, the corrected answer to #3 is used to update the answer of #2, and corrects #2’s answer “Mahesh Bhatt” to “Nanni Moretti”. Through the above process, DeAR is able to help correct wrong reasoning steps and avoid error propagation, which is crucial in enhancing model’s reasoning ability. We will include more detailed cases in the revised version.
>
> **Q4**: Does the method work with stronger LLMs?
>
> **A4**: Thank you for your insightful comments and valuable feedback. In response to your interest, we have conducted further experiments using the GPT-4 backbone to robustly illustrate the effectiveness of our DeAR framework. As indicated in the response to "W2," we present these results to demonstrate the superiority of our approach. On the MATH dataset, a comprehensive benchmark that challenges models with a variety of mathematical reasoning tasks, DeAR has demonstrated superior performance compared to different SOTA methods, including CoT, CoT-SC, ToT, a variant of ToT, and GoT.
>
> **[1]** Self-consistency improves chain of thought reasoning in language models.
>
> **[2]** Large language model guided tree-of-thought.

---

> > ### Comment · Reviewer_dFYv · 2024-08-14
> > **Thanks for your reply**
> >
> > Thanks for your reply. My concerns have been well addressed. I would like to keep my positive score.

---

> ### Author Response · Authors · 2024-08-13
> **Thank You for Any Valuable Feedback**
>
> Thank you for your valuable feedback. We hope that our response adequately addresses your concerns and questions. As our discussion draws to a close, we would appreciate knowing if all your concerns have been resolved. We are open to further discussion if needed.

---

### Author Rebuttal · Authors · 2024-08-06

We sincerely thank all reviewers’ efforts in reviewing our paper. We would like to thank all of them for providing constructive and valuable feedback, which we will leverage to improve this work. We are encouraged by the positive comments from reviewers, including:
- **Motivation**: “offering a fresh perspective on how LLMs can tackle complex problems.” (Reviewer dFYv), “A novel reasoning framework” (Reviewer vfU8), “I find the idea very intuitive, well motivated and mostly well presented” (Reviewer Zsa1)

- **Method**: “novel” (Reviewer dFYv, Reviewer vfU8), “DeAR ensures greater logical consistency compared to traditional methods” (Reviewer dFYv), “The concept of the proposed framework is sound” (Reviewer WGVX), “The entire idea is reasonable and well-aligned with human thinking.” (ANC9), “The structure of the framework is more flexible and reasonable compared to CoT, ToT and GoT” (Reviewer WGVX)

- **Experimental Results**: “ DeAR achieves significant improvements” (Reviewer dFYv), “ the superiority of DeAR over state-of-the-art approaches” (Reviewer  vfU8), “The experiments also demonstrate the method’s versatility” (Reviewer WGVX), “DeAR improves the state-of-the-art and this seems intuitively plausible” (Reviewer Zsa1), “The experiments are sounds” (Reviewer ANC9).

We will specify the detailed responses to all reviewers as follows.

---

### Decision · Program_Chairs · 2024-09-25

**Decision:**

Accept (oral)

**Comment:**

In order to solve intricate reasoning problems, this study introduces the paradigm of  Decompose-Analyze-Rethink (DeAr) cycles for Large Language Models. The key idea is to iteratively build a reasoning tree in a top-down way by breaking the question into sub-question nodes, analyzing them to form rationales, and revisiting answers in parent nodes to refine the reasoning process. This iterative decomposition approach is evaluated experimentally and shows significant improvement over ToT and GoT on several benchmarks.

All reviewers concur that this study makes a significant contribution to solving complex reasoning problems with LLMs. The paper is well-organized, the framework is well-explained, and the comparative experiments are conclusive. In summary, this is a strong piece of work.

While the paper is in good shape, additional explanations about the cycle components could be helpful (Reviewers #ANC9 and #WGVX). It would also be beneficial to incorporate some of the answers to reviewers’ questions into the paper (or the Appendix). Notably, the additional experiments with stronger baselines (Reviewer #dFYv) or more complex tasks (Reviewer #WGVX) would clearly demonstrate the effectiveness of DeAR. In addition, insights from ablation studies (Reviewers #vfU8 and #Zsa1) and the importance of the self-check method (Reviewers #WGVX and #ANC9) could be incorporated.